# An experimental framework to assess biomolecular condensates in bacteria

Y Hoang [1,4], Christopher A. Azaldegui[2,4], Rachel E. Dow[1], Maria Ghalmi[1], Julie S. Biteen [2,3] ✉ & Anthony G. Vecchiarelli [1] ✉

High-resolution imaging of biomolecular condensates in living cells is essential for correlating their properties to those observed through in vitro assays. However, such experiments are limited in bacteria due to resolution limitations. Here we present an experimental framework that probes the formation, reversibility, and dynamics of condensate-forming proteins in *Escherichia coli* as a means to determine the nature of biomolecular condensates in bacteria. We demonstrate that condensates form after passing a threshold concentration, maintain a soluble fraction, dissolve upon shifts in temperature and concentration, and exhibit dynamics consistent with internal rearrangement and exchange between condensed and soluble fractions. We also discover that an established marker for insoluble protein aggregates, IbpA, has different colocalization patterns with bacterial condensates and aggregates, demonstrating its potential applicability as a reporter to differentiate the two in vivo. Overall, this framework provides a generalizable, accessible, and rigorous set of experiments to probe the nature of biomolecular condensates on the submicron scale in bacterial cells.

Though many cellular compartments and organelles use membranes to encapsulate their contents, biomolecular condensates concentrate proteins and nucleic acids without the use of a membrane[1,2]. Condensates have been widely reported in eukaryotes[2,3] and more recently in prokaryotic organisms[4,5], and they have been linked to diverse biological processes important for cell function and physiology[6,7]. Yet, how bacterial condensates maintain their structures, modulate their composition, and regulate internal biochemical reactions is largely unexplored. Our understanding of these questions has improved in eukaryotes[8], but the small size of bacteria has hindered in vivo measurements.

The term condensate is not supposed to prescribe a formation mechanism[2,9]. However, condensate formation has been broadly attributed to phase separation[1,2,7] because of the liquid-like behavior observed in condensates in living cells and in vitro reconstitution. Importantly, condensates that are in a distinct phase from their surrounding environment can be considered as a result of phase separation if they exhibit three key characteristics: a difference in component densities between the two phases, dynamic exchange of components between the two phases, and the formation of a condensate upon reaching a saturation concentration, $c_{sat}$[9]. These observations have been made in eukaryotic cells[10-15] and recently, super-resolution techniques have been implemented to probe sub-diffraction limited bacterial condensates in vivo[16-22]. However, the general use of these approaches remains limited. The growing number of bacterial systems with reported condensates makes it clear that these membraneless organelles play a key role in bacterial cell biology, making a unifying framework to understand their formation across systems necessary. It is therefore imperative that methods become widely available and flexible to study their formation in bacteria.

Here, we present an experimental framework (Fig. 1) to determine whether a bacterial biomolecular condensate forms through phase separation in vivo. We specifically examined Maintenance of carboxysome distribution protein B (McdB) from the cyanobacterium

[1]Department of Molecular, Cellular, and Developmental Biology, University of Michigan, Ann Arbor, MI 48109, USA. [2]Doctoral Program in Chemical Biology, University of Michigan, Ann Arbor, MI 48109, USA. [3]Department of Chemistry, University of Michigan, Ann Arbor, MI 48109, USA. [4]These authors contributed equally: Y Hoang, Christopher A. Azaldegui. ✉e-mail: jsbiteen@umich.edu; ave@umich.edu

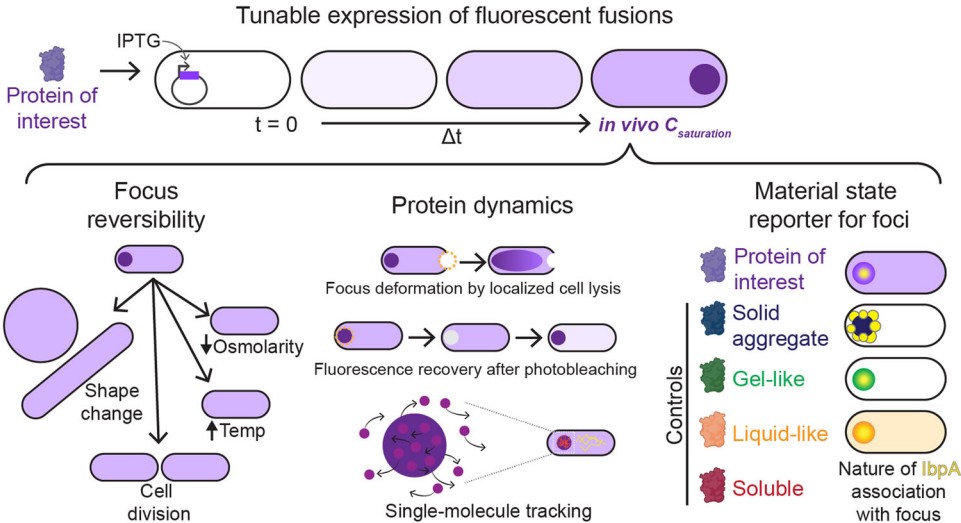

**Fig. 1 | Framework to assess biomolecular condensates in bacteria.** Proteins of interest or control proteins are inducibly expressed to form foci (dark purple circle) in *E. coli* (top). The in vivo saturation concentration ($c_{saturation}$) is quantified. The material state of each focus can be assessed: (*Left*) Condensates dissolve upon cell growth, cell division, cell shape change, temperature shift, or osmolarity shift, whereas insoluble aggregates remain intact. (*Middle*) Analysis of protein dynamics informs on the material properties of the focus. Dashed magenta circles: photobleached areas. (*Right*) The chaperone IbpA (yellow) surrounds insoluble aggregates but penetrates condensates.

*Synecochoccus elongatus* PCC 7942. Carboxysomes are carbon-fixing protein-based organelles that are subcellularly distributed in cyanobacterial cells as well as in some chemoautotrophs. McdB is part of a two-protein system responsible for spatially organizing carboxysomes in the cell[23]. McdB associates with carboxysomes in a currently unknown manner, and acts as an adaptor, linking the carboxysome cargo to its positioning ATPase, called McdA. McdA forms dynamic gradients on the nucleoid in response to McdB-bound carboxysomes, distributing them across the length of the nucleoid. McdB robustly forms phase-separated droplets in vitro, and mutants that are unable to form condensates in vitro are also defective in carboxysome positioning in vivo[23–25]. It remains to be determined if McdB forms condensates in its native host because McdB associates with carboxysomes, making it difficult to parse this association from its potential phase separation activity in vivo. Therefore, we developed and adapted a suite of molecular and cell biology methodologies along with super-resolution imaging techniques in *Escherichia coli* to characterize the formation, solubility, and dynamic exchange of McdB condensates.

*E. coli* as the model system provides a wealth of molecular biology tools. It was also chosen over larger eukaryotic cells, such as yeast, because phase separation is influenced by crowding, and we required a cytoplasm representative of the environment for bacterial proteins. Finally, *E. coli* is also a pragmatic choice for the study of heterologously expressed bacterial proteins because it prevents associations with potential binding partners in the native host. For example, as noted above, in the native organism *S. elongatus*, McdB associates with carboxysomes, which complicates a reductionistic study of McdB phase separation in vivo.

Alongside McdB, we used well-established control proteins that form condensates[26] and insoluble aggregates[27,28]. First, we probed the liquid-like properties of McdB condensates using switch-like expression assays. Next, we used single-cell tunable expression promoters to quantify the conditions for condensate formation and probed condensate reversibility. Moreover, we measured the dynamic exchange of condensate constituents and inferred the material properties of condensates with single-molecule tracking. Finally, we implemented the heat-shock chaperone IbpA as a molecular sensor that differentially associates with condensates and aggregates. Our framework ensures a low barrier to general applicability by users, consolidates an array of assays, and complements widely accessible methods with more advanced techniques such as super-resolution imaging and single-molecule tracking to validate method performance.

## Results
### Probing the liquid-like properties of foci formed by protein overexpression

Bacterial condensates have been mainly studied in vitro, and it is largely unexplored if these proteins form foci with liquid-like properties inside the cell. Overexpression and imaging of fluorescent fusions provide a quick and useful method for gaining insight into the material state of a bacterial focus. Therefore, we first overexpressed a fully functional mNeonGreen-McdB (mNG-McdB) fusion in *E. coli* BL21 cells and performed time-lapse fluorescence microscopy to investigate the behaviors of the McdB protein in vivo. Focus formation was observed in <1 h post-induction with IPTG at 25 °C (Supplementary Fig. 1a and Supplementary Movie 1). DAPI staining of the nucleoid showed that these larger mNG-McdB foci were nucleoid-excluded (Supplementary Fig. 1b). Strikingly, large mNG-McdB foci localized to the inner membrane near sites of local cell curvature (white arrows in Supplementary Fig. 1a and Supplementary Movie 1). Together, we find that overexpressed mNG-McdB forms small foci that eventually become larger nucleoid-excluded structures. These structures appear to wet to the inner membrane and cause local changes in cell morphology.

A hallmark of phase-separated condensates is their responsiveness to changes in the cellular concentration. To probe the concentration dependency of mNG-McdB foci formation in vivo, we treated the cells with A22 which caused a rod-to-sphere transition and corresponding increase in cell volume. We also included a control protein that forms insoluble aggregates in *E. coli*, cI$^{agg}$[28]. The protein cI$^{agg}$ is a truncated and mutated version of the Lambda cI repressor. During this transition, mNG-McdB foci dissolved towards a homogenous phase in the cytoplasm, while the aggregator control mNG-cI$^{agg}$ maintained punctate foci in spherical cells (Supplementary Fig. 1c, Supplementary Movie 2). We analyzed condensation in cells by normalizing all pixel intensities in single cells and calculating the fraction of pixels below a normalized intensity threshold of 50%[16,29]. The condensation coefficients for both McdB and cI$^{agg}$ peaked at about 2 h once fluorescent foci formed (Supplementary Fig. 1d). However, mNG-McdB showed a dramatic decrease in its condensation coefficient

towards a homogenous pixel intensity distribution. In contrast, the condensation coefficient of cI^agg remained high. The results show that McdB redistributed in cells undergoing shape changes, while cI^agg foci increased in size and with minimal fluorescence observed in the cytoplasm.

The drug-induced changes to cell volume were gradual and may have involved pleiotropic effects. Therefore, we developed an approach to probe the dynamic properties of mNG-McdB foci upon an instantaneous change in cell volume. In this localized-lysis method, a high-intensity laser was focused on one cell pole to lyse the cell. Localized lysis caused cell contents to unidirectionally rush out of the cell. Strikingly, upon cell lysis, the McdB focus dispersed towards the opposing open cell pole while the aggregator cI^agg focus remained the same (Supplementary Fig. 1e and Supplementary Movie 3). The observed dynamics of McdB resemble the observed jetting of P granules under sheer stress[30]. Over the 300-second period, the condensation coefficient of McdB decreased by a factor of 2.5, whereas the condensation coefficient of cI^agg remained constant (Supplementary Fig. 1f). Altogether, these results demonstrate that mNG-McdB foci in *E. coli* BL21 exhibit properties consistent with those of condensates.

## Using single-cell tunable expression to probe the formation of biomolecular condensates

In eukaryotic cell biology, the majority of condensate studies perform in vivo measurements using ectopic overexpression[31], analogous to our overexpression assays (Supplementary Fig. 1f). However, phase-separating systems are exceedingly sensitive to changes in concentration, and overexpression may introduce significant caveats in the extrapolation that a protein forms condensates when expressed at lower endogenous levels. We therefore set out to find additional metrics other than overexpression to support claims that a bacterial focus is indeed phase-separated.

We turned to a single-cell tunable expression system to observe condensate formation during controlled protein expression. We fused the fluorescent protein mCherry to the N-terminus of McdB(*mCherry-mcdB*) and placed it under the control of an IPTG-inducible promoter on a pTrc99A expression vector (*pTrc99A-mCherry-mcdB*). Along with our protein of interest, we probed a series of well-established control proteins, all fused to mCherry. We switched from mNG to mCherry for our experiments in MG1665 because we later performed single-particle tracking of our protein set using PAmCherry. Therefore using mCherry for our wide-field fluorescence microscopy, instead of mNG, allowed for a reliable comparison with our single particle data. In addition, later experiments required our protein set to be co-imaged with an *E. coli* chaperone protein fused to sfGFP.

As above, the protein cI^agg is a truncated and mutated version of the Lambda cI repressor that is well-known to form insoluble aggregates in *E. coli*[28]. On the other hand, fluorescent protein fusions to PopTag proteins form condensates via phase separation with tunable material properties that depend on the length of the linker between PopTag and the fusion protein[26]. We engineered two versions of the PopTag fusion: PopTag^SL with a short (six-amino acid) GS repeat linker (*pTrc99A-mCherry-L6-PopTag*) and PopTag^LL with the native linker of the PopZ protein (78-amino acid) (*pTrc99A-mCherry-L78-PopTag*). Based on the linker lengths, PopTag^LL-mCherry condensates should be more fluid than PopTag^SL-mCherry condensates, the latter of which we expect to be more viscous or in a gel-like state[26]. Additionally, we probed a solubilized McdB mutant (McdB^sol), previously shown to be abrogated in its phase separation activity both in vitro and in vivo[25]. Finally, mCherry alone was used as a control for a completely soluble protein. None of the mCherry fusion proteins showed significant cleavage of the mCherry tag compared to mCherry alone when expressed with the pTrc99A promoter (Supplementary Fig. 2). Therefore, mCherry fluorescence served as a reliable measure of protein expression and localization in vivo.

Induced expression of these proteins in *E. coli* cells showed focus formation as a function of increasing protein concentration over time (Fig. 2a-b and Supplementary Fig. 3). Within 1 h of expression, ~80% of cI^agg and 60% of PopTag^SL cells had a focus, while almost no foci were observed in PopTag^LL, McdB, and McdB^sol cells (Fig. 2b). Past 1 h of expression, the percentage of cI^agg and PopTag^SL cells with a focus slightly increased to reach about 90%. PopTag^LL and McdB, on the other hand, showed a sharp increase to 90% and 80% respectively. McdB^sol foci only became evident after 4 h of expression. Strikingly, a notable fraction of the fluorescence signal was localized to the cytoplasm of cells with a PopTag^SL, PopTag^LL, McdB, or McdB^sol focus (Fig. 2a). The focus and the cytoplasmic fraction resemble the dense- and dilute-phases of a two-phase system, respectively. No notable fluorescence was in the cytoplasm of cI^agg cells.

## Probing the in vivo concentration dependence of biomolecular condensates

We next determined the concentration dependence of focus formation among our protein set fused to mCherry. Cells were first categorized based on the presence or absence of a focus. We then plotted the fluorescence concentration of these cell types over the time course of protein expression (Fig. 2c). For cI^agg, cells lacking a focus displayed little to no fluorescence, while cells with a focus displayed a linear increase in concentration over time. The pattern for PopTag^SL was similar to that of cI^agg, consistent with the previously established gel-like nature of PopTag^SL condensates. In contrast, both PopTag^LL and McdB displayed a relatively sharp fluorescence concentration boundary for focus formation. That is, cells without a focus displayed a fluorescence concentration at or below the concentration in cells displaying a focus. McdB^sol foci also exhibited this concentration threshold, albeit foci were only observed at significantly higher concentrations 5 h post-induction.

Together, the data suggest that PopTag^LL and McdB require a threshold concentration for focus formation, consistent with phase separation. To determine their apparent in vivo saturation concentrations ($c_{sat\_app}$), we used quantitative fluorescence microscopy. We first determined the intensity of single mCherry molecules that were spatially isolated prior to photobleaching of fluorescence in the cell. These single-molecule localization images were quantified to measure the number of photons detected per molecule per imaging frame (Supplementary Fig. 4). Next, we calculated the cellular concentration of mCherry-McdB by integrating the total cellular fluorescence intensity per imaging frame and dividing this value by the mCherry single-molecule intensity and the cellular volume. Cells were then classified by the presence or absence of a focus. After a 4-h induction, the average cellular concentration of mCherry-McdB was ~100 μM with no significant difference between cells without a detected focus (average cellular concentration of $98 \pm 9$ μM) and cells with a focus ($113 \pm 8$ μM; Fig. 2d). This observation of condensates in cells with a slightly higher total protein concentration is consistent with McdB undergoing a nearly immediate and concentration-dependent transition to form a focus. We performed the same analysis for cells expressing mCherry-PopTag^LL. The average cellular concentration of this fluid-condensate control was $45 \pm 7$ μM in cells without a focus while cells with a detected focus had an average concentration of $48 \pm 7$ μM (Fig. 2d). Together, the results provide a $c_{sat\_app}$ at which condensates form; for mCherry-PopTag^LL and mCherry-McdB, we estimate the $c_{sat\_app}$ to be between 38 – 55 μM and 89 – 121 μM, respectively.

Full-length PopZ has a similar linker length as our PopTag^LL fusion. In vitro, PopZ forms condensates at concentrations as low as 1 μM[26]. However, PopZ micro domains in its native host, *Caulobacter crescentus*, were estimated to be ~ 50 μM at the cell poles. Therefore, our in vivo $c_{sat\_app}$ of PopTag^LL is consistent with previously reported in vivo values[17,26]. As for McdB, its cellular concentration in its native

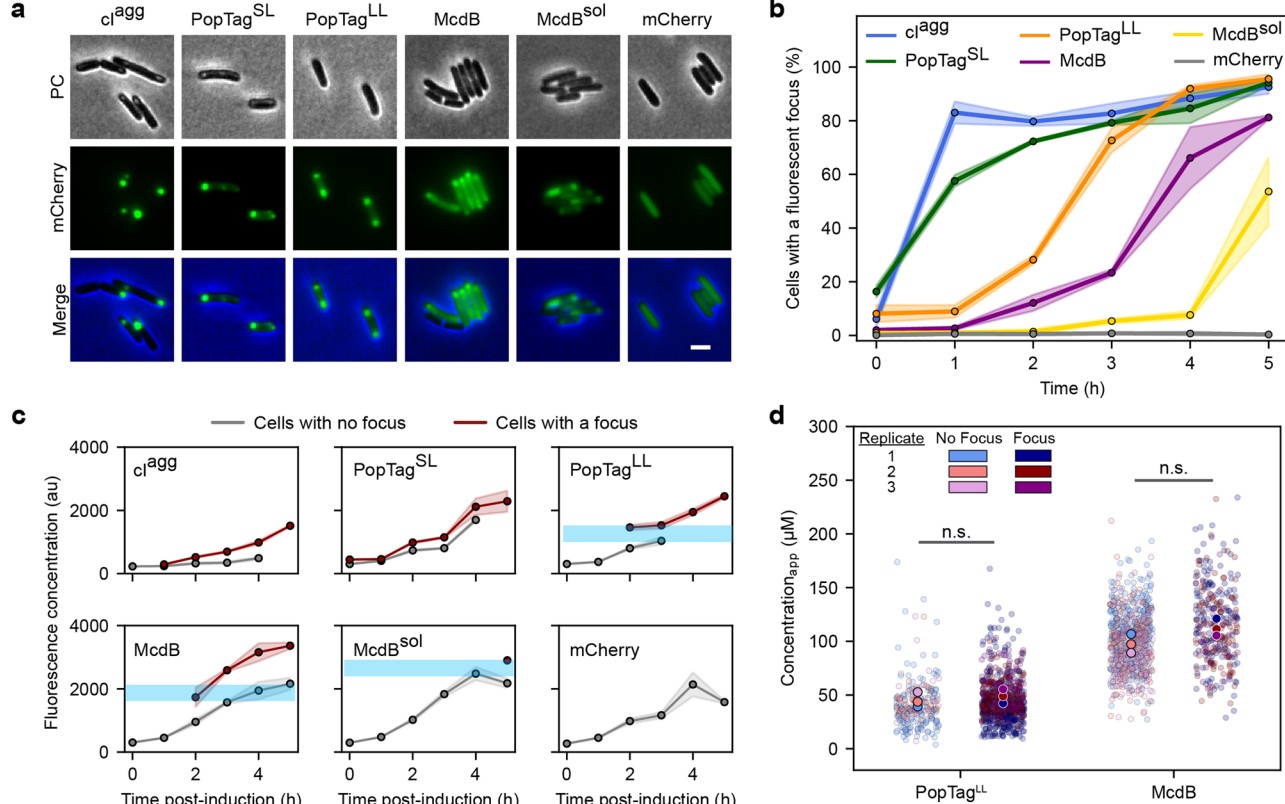

**Fig. 2 | Proteins that phase separate in vitro form foci above a quantifiable in vivo $c_{sat}$. a** Representative images of mCherry fusion protein foci in *E. coli* at 4 h post-induction. Phase contrast (PC) (gray), mCherry channel (green), and merged images are shown. Images are representative of three biological replicates. Scale bar: 2 μm. **b** Percent of cells with a focus over induction time. Solid lines and shading represent the average and standard deviation, respectively, over three biological replicates. $N > 100$ cells for each protein at each time point per replicate. **c** Fluorescence concentrations of proteins at times post-induction in cells from (**b**). Cells were classified for the presence of a focus as in (**b**). Solid lines and shading represent the average and standard deviation, respectively, over three biological replicates. $N > 100$ cells for each protein at each time point per replicate. Each data point is shown if it totals at least 10% of the population at its corresponding time point. Light blue shading is added to highlight potential saturation concentration. **d** Quantification of an apparent cellular $c_{sat}$. Cells are classified by the presence or absence of a focus. Data points correspond to individual cells. $N > 100$ cells for each protein per replicate. Two-sided Welch's *t*-test was done on the mean of the replicates. n.s. indicates no statistically significant difference between the samples. Source data are provided as a Source Data file.

host, *Synechococcus elongatus*, has yet to be determined. In vitro, we have shown that McdB can form condensates at concentrations as low as 2 μM[25]. Here, we report estimates several fold higher than the in vitro $c_{sat}$. However, given that McdB associates with carboxysomes in *S. elongatus*, we speculate that this association significantly increases the local concentration of McdB and/or drops the $c_{sat}$ of McdB, which likely explains this discrepancy.

**Condensates coexist with a soluble phase**

A hallmark of phase separation is the existence of a soluble fraction in the cytoplasm. To quantify the ratio of mCherry protein fusion concentration in the focus (dense phase) to the concentration in the cytoplasmic fraction (dilute phase), the partitioning of each fusion was measured (Fig. 3a). The largest partitioning was measured for the aggregator control, cl[agg], in which there was no detectable fluorescent signal in the cytoplasm. The PopTag[SL] condensate control partitioned to a greater extent than the more fluid condensate control PopTag[LL]. The partitioning of our protein of interest, McdB, was intermediate relative to the PopTags. The condensation coefficients determined by using the same dataset for each protein (Fig. 3b; Supplementary Fig. 5) were consistent with the partition ratios (Fig. 3a).

To further inspect the partitioning of proteins to polar foci, we induced photoactivatable (PA) mCherry fusions of each protein and performed single-molecule localization super-resolution microscopy to generate normalized localization density heat maps (Fig. 3c). Consistent

with our previous results, cl[agg] had minimal localization density in the cytoplasm under all conditions, which suggests nearly all protein is recruited to the polar aggregate. On the other hand, the polar density of PopTag[SL] increased with increasing protein concentration, but these cells also maintained a protein fraction in the cytoplasm. Strikingly, PopTag[LL] and McdB displayed a transition between no induction and 2 h post-induction, indicative of focus formation. Intriguingly, McdB[sol], which does not form condensates in vitro, formed high-density regions at the poles; consistent with bulk fluorescence measurements (Fig. 2a). To abrogate phase separation in this McdB[sol] mutant, the net negative charge of its IDR was increased[25]. Therefore, we speculate that the localization pattern of the fluidized McdB[sol] condensate is due to nucleoid exclusion by repulsive electrostatic interactions. Consistently, we found that the polar McdB[sol] signal encroached throughout the cytoplasm as the nucleoid was compacted via drug treatment (Supplementary Fig. 6 and Supplementary Movie 5).

Collectively, our results demonstrate the use of a tunable expression system that shows the concentration-dependent formation of condensates and exhibits a two-phase behavior, indicative of phase separation. Phase separation theory predicts that while many small condensates form at the initiation of phase separation, this number decreases, and the sizes of those that persist increase via coalescence[32,33]. Our findings here are consistent with the expected final ground state being a single large condensate that coexists with a dilute phase. Some cells had foci at opposing poles (see

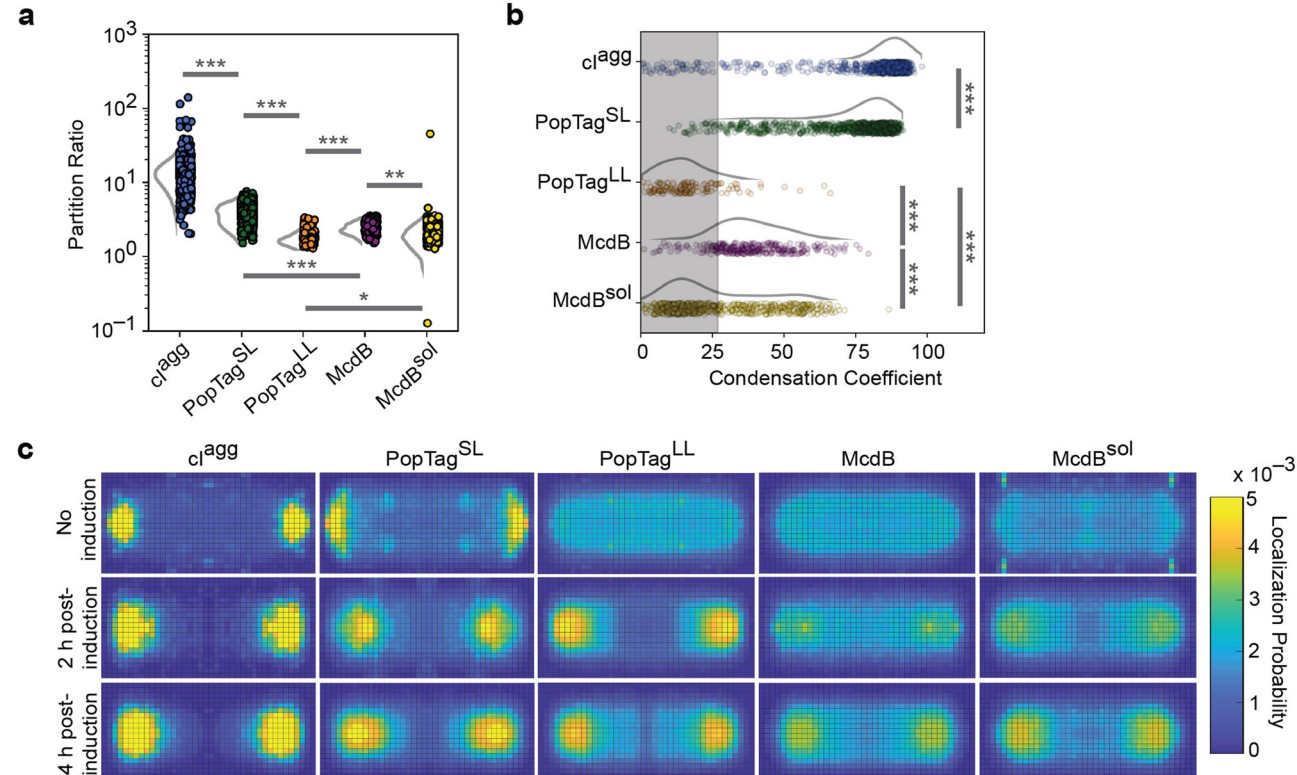

**Fig. 3 | Cells that form condensates maintain a soluble cytoplasmic fraction.**
**a** Partition ratios for specified mCherry fusions with detected foci (cI$^{agg}$, $N = 995$ cells; PopTag$^{SL}$, $N = 939$ cells; PopTag$^{LL}$, $N = 147$ cells; McdB, $N = 341$ cells; McdB$^{sol}$, $N = 714$ cells). **b** Single-cell condensation coefficients for specified mCherry fusions with detected foci cI$^{agg}$, $N = 1019$ cells; PopTag$^{SL}$, $N = 942$ cells; PopTag$^{LL}$, $N = 151$ cells; McdB, $N = 343$ cells; McdB$^{sol}$, $N = 717$ cells. An intensity threshold $I = 0.3$ was used. Data points correspond to individual cells and were normalized the mean condensation coefficient of cells expressing mCherry only. The shaded region

represents the measurement range for cells expressing a uniform mCherry signal. The curves next to the scatter plots were obtained via kernel density estimation. $N$ values are the same as in (**a**). **c** Normalized single-molecule localization 2D histograms. Single-molecule localizations of specified PAmCherry fusion proteins were collected, projected, and binned onto a normalized cell shape to generate the heat map. Localizations were collected from 30 cells or more per condition over three biological replicates. *$p < 0.05$, **$p < 0.01$, and ***$p < 0.001$ by two-sided Welch's $t$-test. Source data are provided as a Source Data file.

---

Supplementary Fig. 1B). However, this can be attributed to the steric effects of nucleoid exclusion that prevent condensates from physically interacting and coalescing.

**Changes in cellular concentration, osmolarity, and temperature probe the reversibility of condensates**

A hallmark of condensates is reversibility in response to changes in protein concentration and chemical environment. We implemented several approaches to probe condensate reversibility of our protein set in *E. coli* cells. First, we set out to determine if dropping protein concentration below $c_{sat}$ would dissolve condensates, thus demonstrating solubility as a driving force for condensate formation (Fig. 4a-f). Each protein was expressed via IPTG induction, and once foci formed, the inducer was removed from the media. Cells were then allowed to grow and divide, and in doing so, dilute the concentration of the mCherry fusion proteins. cI$^{agg}$ foci remained at the pole from which they formed even as cells grew and divided over 10 h ($\geq 6$ generations) (Fig. 4a-b and Supplementary Movie 6). This result shows no threshold concentration dependence in the formation and maintenance of cI$^{agg}$ foci, and further supports its insolubility previously reported[28].

In contrast, PopTag$^{SL}$, PopTag$^{LL}$, and McdB foci all exhibited concentration-dependent dissolution. Roughly 75% of PopTag$^{SL}$ foci dissolved and had an average lifespan of $1.6 \pm 2$ h (Fig. 4a-b, Supplementary Fig. 7a and Table 1). The generational dilution effect was more immediate with PopTag$^{LL}$ (Supplementary Fig. 7a), where all foci dissolved within $0.3 \pm 0.2$ h or roughly a single cell division event (Fig. 4b and Table 1). When compared to these controls, McdB focus

dissolution mirrored that of the fluid PopTag$^{LL}$ condensate, with ~ 97% of foci dissolving within $1.4 \pm 1$ h or within one to three cell divisions (Fig. 4a-b and Supplementary Table 1). Indeed, most McdB foci dissolved following a single cell division event (Supplementary Movie 6). Furthermore, we measured the fluorescence concentration of cells that exhibited focus dissolution at the frames immediately preceding and immediately following the dissolution event (Fig. 4c, f). Consistent with our previous $c_{sat\_app}$ measurements, the average fluorescence concentration of McdB when foci dissolved was two-fold higher than cells expressing PopTag$^{LL}$ (Fig. 4c). PopTag$^{LL}$ having a higher fluorescence concentration than PopTag$^{SL}$ when foci dissolved is also consistent with the longer linker length increasing $c_{sat}$[26].

Intriguingly, some cells reformed McdB and PopTag$^{SL}$ foci immediately after cell division (Fig. 4g-j). We hypothesized that this phenomenon was due to an asymmetric inheritance of protein upon cell division, which drove the concentration in one daughter cell above $c_{sat}$. Analysis of several division events for cells expressing McdB and PopTag$^{SL}$ (see Methods) confirmed our hypothesis: most daughter cells that form foci have inherited more fluorescence signal than the other corresponding daughter cell (Fig. 4h, j).

We also probed condensate solubility by diluting protein concentration via an increase in cell length without cell division or further protein expression. *E. coli* cells were prepared as described above, but with the inclusion of the cell division inhibitor cephalexin. We expected similar trends for dilution via cell elongation as in cell division. Indeed all cI$^{agg}$ foci were observed to persist throughout the experiment duration ($> 7.5$ h) (Fig. 4d-e and Supplementary Movie 7). Also

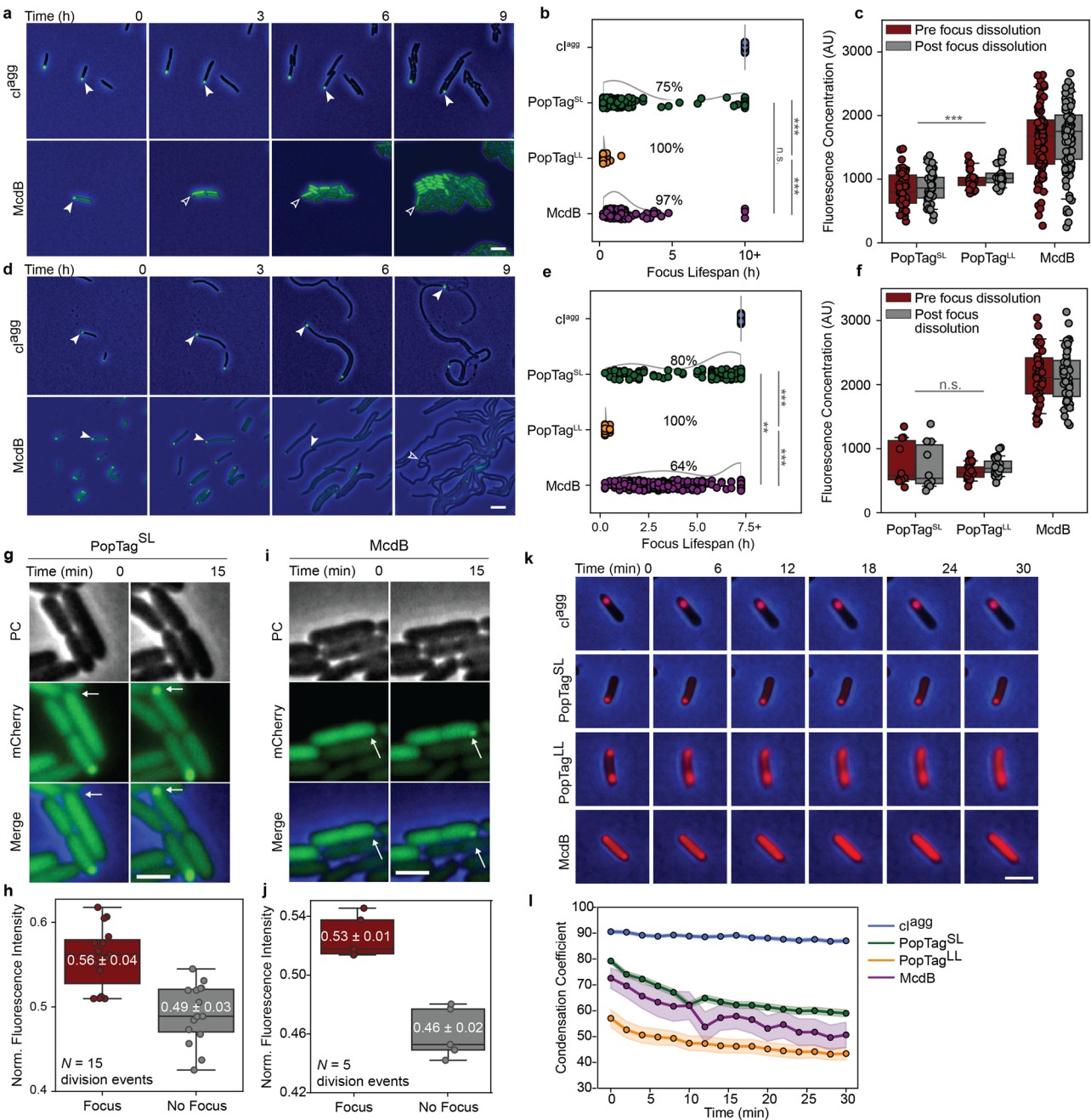

**Fig. 4 | Changes in cell volume and temperature drive condensate reversibility.** The phase contrast (PC) channel is shown in blue or gray and the mCherry channel is in green. Sample size is shown in Supplementary Table 1. Images are representative of four biological replicates. Scale bars: 1 μm (**a**). Generational-dilution dissolves condensates. White arrows demarcate the cellular location of the same focus over time. Blank arrows demarcate the same cellular position now absent of a focus. **b** Lifespan of protein foci during cell division. Data points correspond to individual foci. The curves next to the scatter plots are obtained via kernel density estimation. Percentages indicate the percent of foci that dissolved prior to the last frame in the data collection. Statistical analysis was performed on the lifespans of foci that dissolved. **c** Fluorescence concentration of cells in the frames pre- (red) and post- (gray) the disappearance of a fluorescent focus. **d**. Cell elongation dissolves condensates. Same as in (**a**). **e** Lifespan of protein foci during cell elongation.

Same as in (**b**). **f** Fluorescence concentration of cells in the frames pre- (red) and post- (gray) the disappearance of a fluorescent focus. Same as in (**c**). **g** Focus reformation after cell division. White arrows demarcate the same cellular location before and after cell division. **h** Quantification of daughter cells in (**g**). **j**–**l** Same as in (**g**, **h**). **k** Effects of temperature shift on focus stability. **j** Quantification of (**k**). Condensation coefficient of cells over time using an intensity threshold $I = 0.5$. Colored solid lines and shading represent the average and 95% confidence interval, respectively, of at least 7 cells per sample at every time point. ***$p < 0.001$, **$p < 0.01$, n.s. indicates no statistically significant difference between the samples by two-sided Welch's t-test. All box plots presented show the median, upper and lower quartiles, and the whiskers represent the 1.5 × interquartile range. Source data are provided as a Source Data file.

consistent with the generational dilution experiment, ~ 80% of PopTag$^{SL}$ foci dissolved during cell elongation; albeit with a slightly longer average focus lifespan ($4.3 ± 2.3$ h) (Fig. 4e, Supplementary Fig. 7b and Table 1). The more fluid PopTag control, PopTag$^{LL}$, also

readily dissolved with an average lifespan of $0.3 ± 0.1$ h. Like PopTag$^{SL}$, McdB foci also exhibited an extended lifespan of $3.2 ± 1.6$ h, compared to the generational dilution experiment. Moreover, only 64% of foci dissolved−a significant decrease compared to the generational

dilution experiment. The reason for condensate persistence during cell elongation and not during cell division is unknown. However, we speculate that the altered cytoplasmic conditions (e.g. effective volume and crowding) of multi-nucleate elongated cells, compared to cells undergoing vegetative growth, play a role in the observed differences in focus dissolution. Consistent with this hypothesis, the fluorescence concentration of cells expressing PopTag$^{LL}$ and McdB when foci dissolved during cell elongation (Fig. 4f) were shifted when compared to the generational dilution experiment (Fig. 4c).

Hyperosmotic stress is commonly used to induce condensate formation in eukaryotic cells, via increases in the concentration of phase-separating proteins as well as the macromolecular crowding and the ionic strength of the cell. In bacteria, condensate formation and dissolution in response to changes in osmolarity have yet to be described to the best of our knowledge. We expressed each protein fused to mCherry, and once foci formed, the inducer was removed. Cells were then osmotically stressed at 300 mM NaCl for 15 min and allowed to recover for 15 min or 30 min prior to imaging (Supplementary Fig. 7c-d). Immediately following osmotic upshift, cI$^{agg}$ cells displayed slightly more foci, compared to untreated cells (Supplementary Fig. 7c-d). Upon recovery, the foci number declined over time, but at a rate that matched untreated cells, showing this decrease was not due to a recovery from the osmotic upshift. On the other hand, cells expressing PopTag$^{SL}$, PopTag$^{LL}$, or McdB, all formed significantly more foci immediately following osmotic upshift, and upon recovery, foci count rapidly decreased to the level of untreated cells (Supplementary Fig. 7d). The findings suggest that McdB and PopTag condensates are responsive to changes in osmolarity.

Finally, temperature is also a well-established modulator of protein phase separation. We hypothesized that increasing the temperature of cells with a preformed focus would influence condensates, while having little to no effect on insoluble aggregates. Cells expressing the mCherry fusions were induced to form foci, placed on ice to further cool the cells, and then imaged on a stage-top incubator set at 37 °C. As expected, cI$^{agg}$ foci remained unchanged throughout the temperature increase (Fig. 4k-l). PopTag$^{SL}$ foci did not readily dissolve, but cells exhibited a gradual increase in the cytoplasmic fluorescence intensity. Consistently, the single-cell condensation coefficient for PopTag$^{SL}$ decreased over the temperature shift, indicating a transition in protein distribution in the cell (Fig. 4l). PopTag$^{LL}$ and McdB foci dissolved more readily towards a homogenous distribution (condensation coefficient ≈ 50) (Fig. 4l). These results demonstrate that, unlike insoluble aggregates, phase-separated condensates are sensitive to temperature change, and the extent of responsiveness may correlate with the material properties of the condensate.

Together, using a series of reversal assays that probe the influences of protein concentration, osmolarity, and temperature, we observed focus dissolution and a corresponding turnover of protein into the cytoplasmic phase. The findings are consistent with the reversal to a one-phase system once the concentration has decreased below $c_{sat}$. These characteristics further support condensate formation by the PopTag fusions via phase separation and suggest a similar formation process for McdB condensates.

### Probing the dynamic rearrangement, confinement, and exchange of biomolecular condensates

To probe the dynamic rearrangement of molecules within a focus and their exchange with the surrounding cytoplasm, we first implemented fluorescence recovery after photobleaching (FRAP) on the mCherry fusion foci. The aggregator control, cI$^{agg}$, exhibited no fluorescence recovery (Fig. 5a-b), supporting the static nature of the proteins within these aggregates and the absence of protein exchange with the cytoplasm. The PopTag fusions partially recovered to an extent consistent with their respective fluidity levels: PopTag$^{SL}$ and PopTag$^{LL}$ recovered to ~10% and 40%, respectively, of the initial fluorescence intensity (Fig. 5b).

Recovery of McdB foci was once again similar to that of the fluid PopTag$^{LL}$ condensate. Taken together, the data suggest that McdB exhibits dynamics similar to that of the fluid PopTag$^{LL}$, while the minimal recovery of PopTag$^{SL}$ is consistent with its predicted gel-like state[26]. Given the limited number of pixels that make up the photobleached foci, we note that the fluorescence recovery contributions from internal rearrangements of molecules within a focus cannot be distinguished from the exchange of molecules with the surrounding milieu[31]. While FRAP is commonly used to determine if a compartment is liquid-like, this method is solely a measure of exchange dynamics, so FRAP alone is not a reliable "gold standard" measure of the material state of a focus.

Therefore, to further investigate the dynamics of our set of proteins both in the focus and in the cytoplasm, we implemented single-molecule localization microscopy and tracked the movement of individual molecules. We then determined the apparent diffusion coefficients ($D_{app}$) of the PAmCherry fusions to the screened proteins under different induction conditions by measuring the mean square displacement (MSD) of individual trajectories as a function of time lag, $\tau$. Prior to induction, low-level leaky expression of all fusions displayed a fast diffusive state ($D_{app,fast}$) (Supplementary Fig. 8 and Table 2). Trajectory mapping back onto the cell showed the $D_{app,fast}$ population corresponds to free diffusion in the cytoplasm (Fig. 5c). cI$^{agg}$ and PopTag$^{SL}$ also displayed a small population of nearly static molecules ($D_{app,slow}$) (Supplementary Fig. 8 and Table 2), with trajectories that mapped to the cell poles (Fig. 5c). Upon induction, the near-static fraction of molecules increased for cI$^{agg}$ and PopTag$^{SL}$, and became the dominant state for cI$^{agg}$ (Supplementary Fig. 8 and Table 2). A diffuse state, slower than free diffusion, also emerged for McdB, which mapped to the cell poles (Fig. 4c). The vast majority of McdB$^{sol}$ molecules, on the other hand, remained in the fast diffusive state but remained nucleoid excluded (Fig. 4c); consistent with our wide-field microscopy of this fluidized McdB mutant (Supplementary Fig. 6). PopTag$^{LL}$ also exhibited nucleoid exclusion (Fig. 5c), despite remaining highly mobile. As stated earlier, we speculate that this behavior is caused by charge repulsion with the nucleoid.

By comparing the average mobility of $D_{app,slow}$ populations across all fusions, we find that cI$^{agg}$ molecules within a focus were essentially static (Fig. 5d), consistent with being an aggregate. PopTag$^{SL}$ molecules in foci displayed an intermediate mobility consistent with a gel or highly-viscous fluid state. All PopTag$^{LL}$ molecules, on the other hand, essentially displayed a monomodal diffusive distribution across all induction conditions, consistent with this version of the PopTag being highly fluid. Finally, McdB displayed a $D_{app,slow}$ state only slightly less mobile than the $D_{app,fast}$ population, consistent with a liquid-like condensate. Although our fluidized mutant, McdB$^{sol}$, displayed a $D_{app,slow}$ population similar to that of McdB, $D_{app,slow}$ was a minor fraction of the total population (Fig. 5e and Supplementary Fig. 8).

A recent study that reports on the inhomogeneity of the bacterial cytoplasm finds slowed apparent protein diffusion at the cell poles compared to other regions of the *E. coli* cell[34]. The finding highlights the importance of comparative studies of in-focus diffusion measurements for foci occupying the same cellular location. It is likely that slowed protein diffusion contributes to condensate formation at cell poles. But, even if diffusion is slowed at the poles to some extent, our in-focus diffusion measurements are much slower and non-uniform across our protein set; supporting the conclusion that interactions involved in condensate formation further affect protein diffusion. Collectively, the data parse the dynamic exchange of molecules in foci and the surrounding cytoplasm, and delineate the spectrum of possible mobilities within a focus, thus allowing for inference of its material state.

### IbpA as a reporter to differentiate between condensates and aggregates

The heat shock chaperone, IbpA, has been shown to colocalize with insoluble aggregates in *E. coli*[28]. We therefore initially thought that IbpA

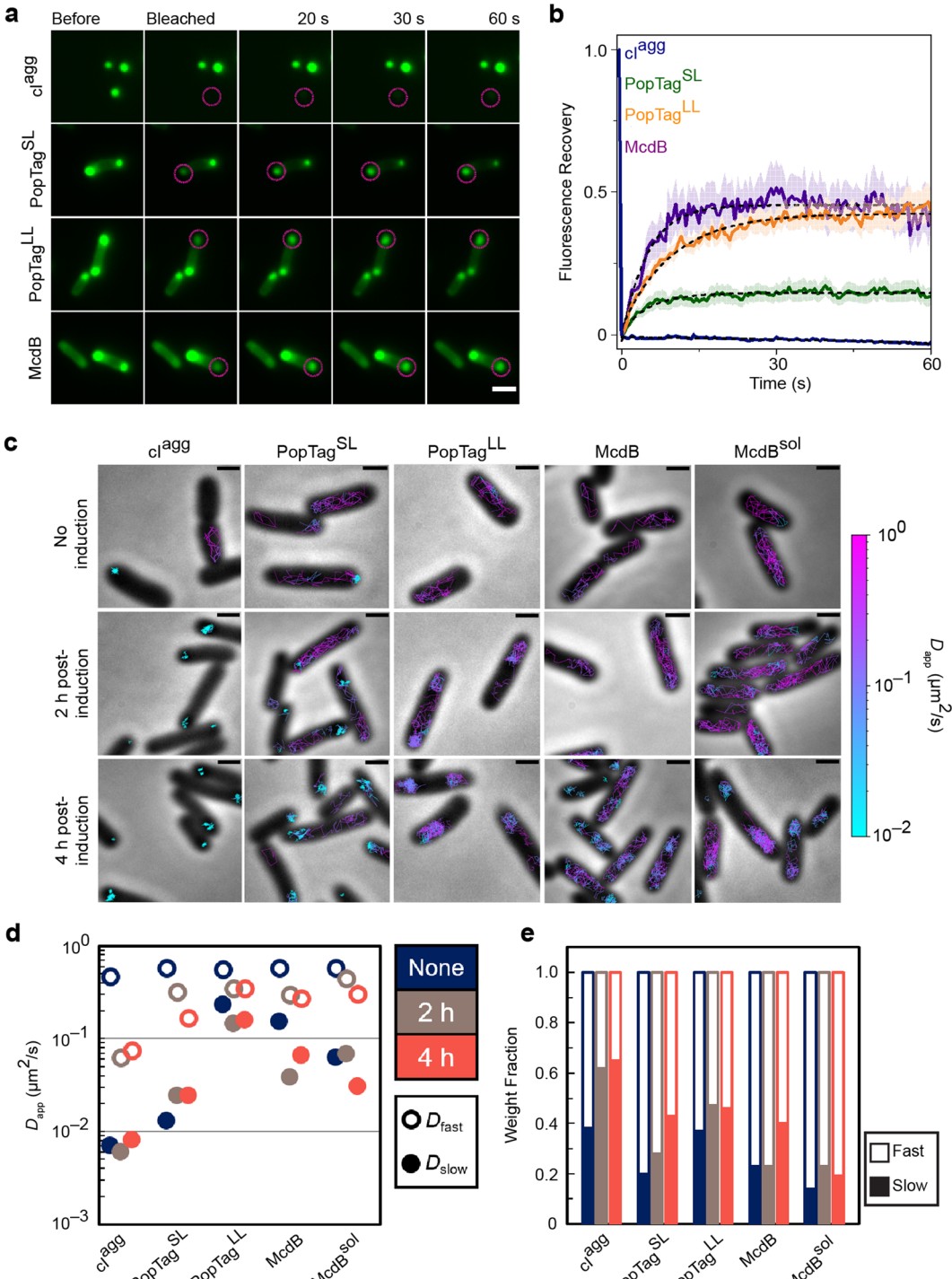

**Fig. 5 | FRAP in concert with single-particle tracking informs on the material state of foci in bacterial cells. a** Fluorescence recovery of mCherry-fusion proteins after photobleaching. Magenta circles indicate the FRAP region. Scale bar: 2 μm. **b** Quantification of fluorescence recovery of mCherry fusion proteins. Shading represents the standard error of the mean. cl$^{agg}$: $n = 14$ foci; PopTag$^{SL}$: $n = 12$ foci; PopTag$^{LL}$: $n = 14$ foci; McdB: $n = 10$ foci. **c** Representative single-molecule trajectories. Overlay of tracks, color-coded according to the track apparent diffusion coefficient, obtained from representative *E. coli* cells expressing PAmCherry fusion proteins. Scale bars: 1 μm. **d** Diffusion

coefficients of PAmCherry fusion proteins. The diffusion coefficients of the indicated PAmCherry fusions were obtained by fitting the histograms of the log diffusion coefficients of single tracks to a two-component Gaussian mixture model assuming that the fusion proteins are composed of a slow (closed circles) and fast (open circles) fraction. **e** Weight fractions of mobility states of PAmCherry fusion proteins. Two-component Gaussian mixture fitting results show an increase in slow mobility fraction (solid bars) and a decrease in fast mobility fraction (empty bars) as protein concentration increases. Source data are provided as a Source Data file.

may work as an in vivo reporter that discriminates between aggregates and condensates, colocalizing with the aggregator control and not recognizing PopTag- or McdB-condensates. We expressed the mCherry fusion proteins under focus-forming conditions in an *E. coli* strain that

expressed a chromosomal fluorescent reporter of IbpA (IbpA-msfGFP) and then observed colocalization patterns of the mCherry fusion proteins with IbpA. Unexpectedly, the IbpA foci strongly colocalized with all proteins surveyed (Supplementary Fig. 9a), showing that IbpA does

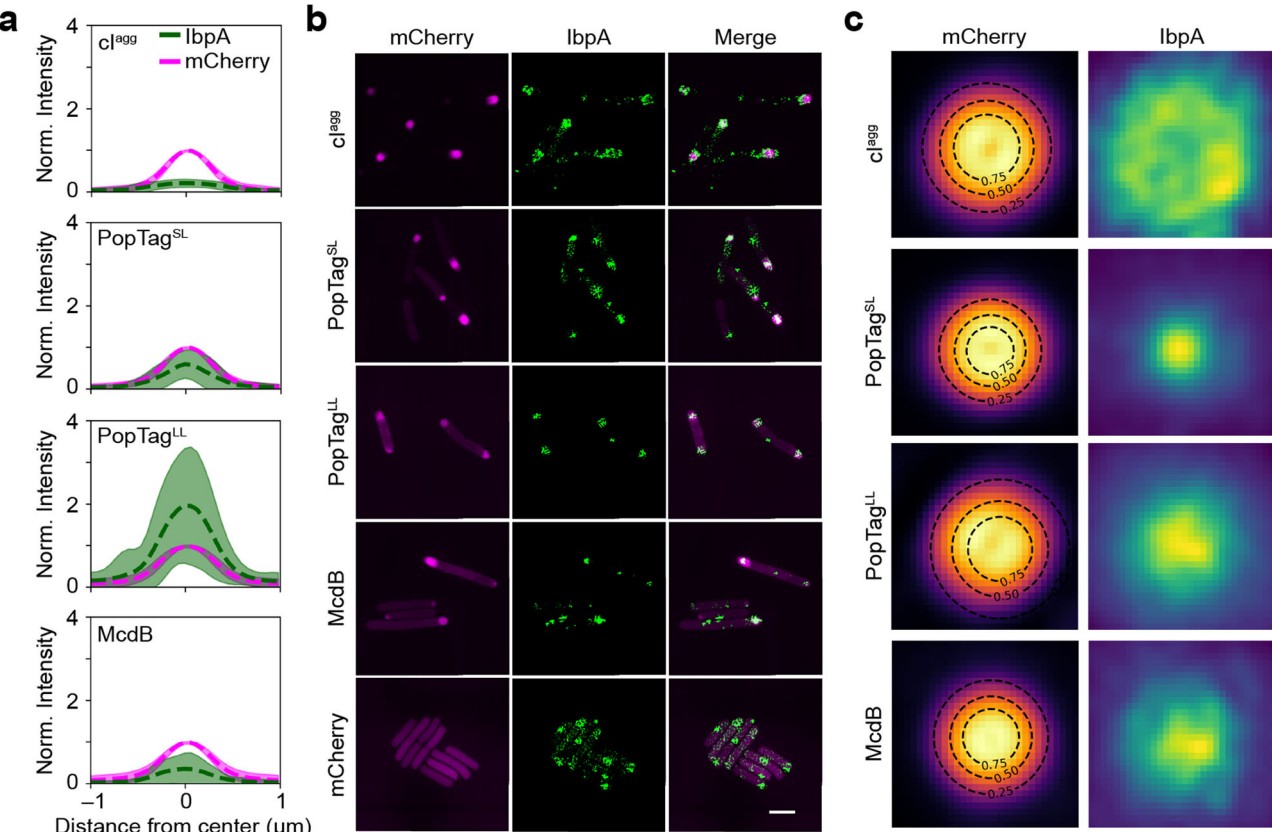

**Fig. 6 | Nature of IbpA association with mCherry fusion foci by 2D SIM.**
**a** Fluorescence intensity profile lines of mCherry fusion foci (magenta) and IbpA-msfGFP foci (green). Dashed lines and shading are the mean and standard deviation, respectively, of fluorescence intensities normalized to the mCherry signal. $cl^{agg}$: $n = 348$ foci; PopTag$^{SL}$: $n = 540$ foci; PopTag$^{LL}$: $n = 547$; McdB: $n = 295$ foci. **b** 2D SIM images of mCherry fusion proteins foci (magenta) and IbpA foci (green) are

shown. Images are representative of three biological replicates. Scale bar: 2 μm.
**c** Fluorescence projection images of mCherry fusion foci and local IbpA-msfGFP signal. Black dashed lines indicate the contour at 0.25, 0.50, and 0.75 of the 2D Gaussian fit amplitude surrounding the mCherry fusion focus. $cl^{agg}$: $n = 84$ foci; PopTag$^{SL}$: $n = 506$ foci; PopTag$^{LL}$: $n = 457$ foci; McdB: $n = 217$ foci. Source data are provided as a Source Data file.

not specifically localize to misfolded protein aggregates in the cell as ascribed by current models[28]. Upon further investigation, we observed that the localization pattern of IbpA association with the aggregate control versus the condensates was quantifiably different. We also found that IbpA appeared to coat, as opposed to penetrate, the $cl^{agg}$ foci (Supplementary Fig. 9a). Consistently, we quantified and compared the diameters of the IbpA signal projections relative to each mCherry fusion (Supplementary Fig. 9b) and found that the IbpA signal was spread over a larger area than the $cl^{agg}$ foci.

We extracted from the wide-field fluorescence data an intensity profile line plot of foci in both color channels. When normalized to the maximum intensity of the mCherry signal, we observed significant differences in the relative amounts of IbpA penetrating each of the mCherry fusion foci (Fig. 6a). On average, the max intensity of IbpA was $20 \pm 9$ % of that of $cl^{agg}$ foci, while it was $58 \pm 33$, $188 \pm 131$, and $33 \pm 35$ % for PopTag$^{SL}$, PopTag$^{LL}$, and McdB foci, respectively. The limited amount of IbpA relative to $cl^{agg}$ suggests that while IbpA can sense the aggregate, it cannot penetrate it well. In contrast, the higher amounts of IbpA in all other foci pointed to its ability to penetrate more fluid assemblies. This result demonstrates varying degrees of IbpA penetration within the corresponding mCherry fusion foci, which correlate well with their hypothesized material state. Together, the findings suggest a different pattern of colocalization of IbpA with the examined biomolecular condensates and aggregates.

To better resolve the colocalization patterns between IbpA and the mCherry fusion foci, we imaged cells under the same conditions as described earlier by structured illumination microscopy (SIM). With

the increased resolution in both channels, the localization patterns of IbpA relative to the mCherry fusion foci were more easily observed (Fig. 6b). We registered the detected mCherry fusion protein foci at their center positions, normalized the intensities, and projected all detected focus images to produce an average image of each mCherry fusion protein focus. We observed that IbpA forms a rosette pattern around the mCherry-$cl^{agg}$ foci, a punctate focus within mCherry-PopTag$^{SL}$ foci, and an amorphous focus pattern with mCherry-PopTag$^{LL}$ and mCherry-McdB (Fig. 6c). This result demonstrates that IbpA exhibits a different colocalization pattern with biomolecular condensates compared to aggregates and better penetrates more fluid condensates. These patterns serve as a proof of concept that IbpA serves as a reporter capable of differentiating between these macromolecular assemblies in vivo. Future studies will determine if other chaperones share this activity.

## Discussion

In this study, we developed an experimental framework to assess the material state of fluorescent foci in bacteria; specifically, whether a focus can be described as a phase-separated condensate. When studying a new condensate, it is crucial to first determine the conditions under which it forms or dissolves. Inducer-controlled protein expression combined with quantitative fluorescence microscopy demonstrated general applicability in identifying the in vivo $c_{sat}$ for condensate formation. Our results also indicate that in vivo $c_{sat}$ may vary within the cell population. Another critical metric is to determine the reversibility of a focus, as it is one of the hallmarks that distinguish

condensates from aggregates. Our approaches for decreasing cellular protein levels below $c_{sat}$, via changes in cell shape, osmolarity, temperature, or generational dilution, provide accessible methods to probe the reversibility of condensate formation.

The dynamics of proteins inside a focus should then be determined by FRAP and single-molecule tracking. While FRAP is more accessible, it is challenging to collect data from a large number of cells. Single-molecule tracking experiments, on the other hand, enable the assessment of a large sample size and more in-depth analysis of protein dynamics within both the cytoplasm and the focus. However, the material state of a focus cannot be determined based on the diffusion coefficient alone. Even when the diffusion coefficient is relatively low, focus formation can still be reversible, as observed with the gel-like control protein in our study. Therefore, a combination of reversibility and dynamic assessments is essential to determine the material state of a focus in bacterial cells.

Finally, we identified the heat-shock chaperone, IbpA, as a molecular sensor that surrounds solid aggregates but penetrates condensates. We demonstrated the versatility of these approaches by using focus-forming control proteins that span the spectrum of material states from a solid aggregate to a highly fluid condensate. When compared to these control proteins, we find that our protein of interest, McdB, robustly phase-separates into liquid-like condensates in vivo. As shown with the fluidized mutant of McdB, this framework can also be combined with mutagenesis studies to determine the regions and residues of a protein governing its material state and phase separation behavior in a bacterial cell.

Our framework overcomes current limitations in identifying the material state of a fluorescent focus and can be used to assess the phase separation activity of expressed recombinant proteins and bacterial inclusion bodies (IBs). IBs are mesoscale protein aggregates, once strictly proposed as being composed of nonfunctional and misfolded proteins[35]. Phase separation has only recently been considered in the assembly and organization of IBs[36]. For example, several condensates have been shown to mature into gels and solid amyloids[1]. However, direct determination of whether a bacterial IB is a liquid, gel, solid, or a mixture of these states remains to be demonstrated.

Inclusion body binding protein A (IbpA) of E. coli belongs to the conserved family of ATP-independent small heat shock proteins, well-established in binding protein aggregates and driving them towards reactivation-prone assemblies[37–39]. As such, we presumed IbpA would serve as a molecular sensor that would selectively associate with protein aggregates, but not condensates. Instead, we found that IbpA surrounded protein aggregates and penetrated condensates. Moreover, the degree to which IbpA colocalized with condensates strongly correlated with increasing fluidity. Consistent with our findings, Yoo et al. recently showed that condensates are dispersed by chaperones far more rapidly than misfolded aggregates[40]. These findings warrant a reevaluation of the function of chaperone systems governing protein homeostasis and demonstrate the utility of IbpA, and potentially other chaperones, as molecular sensors for the material state of fluorescent foci in bacteria.

The framework was built by contrasting the principles of aggregation versus phase separation behaviors of proteins. While not an exhaustive list, the approaches used here are highly accessible and probe several aspects of condensate assembly, maintenance, and dissolution. The framework is not without limitations. First, the approaches used here do not provide mechanistic insights into the process of phase separation, that is, whether condensate formation is driven purely by liquid-liquid phase separation (LLPS), phase separation coupled to percolation (PSCP), or phase separation coupled to other phase transitions (PS + +)[9]. The current framework also does not include an assessment of the boundary between the dense and dilute phases, which is important in understanding the finite interfacial tension that hinders macromolecular transport across the boundary.

Third, for broad use and accessibility, we developed the experimental framework using heterologous protein expression in E. coli, which may not accurately reflect the in vivo conditions of the native host. Finally, low-level leaky expression of inducible promoters may preclude the examination of proteins that phase separate at very low in vivo $c_{sat}$. Despite these limitations, this framework provides broad-use and systematic approaches that address ongoing debates over the rigor and standardization of phase separation assessments in bacterial cells.

## Methods

### Bacterial strains, plasmids, and primers
Strains, plasmids, and primers used in this study are listed in Supplementary Tables 3 and 4. All constructs were made using Gibson assembly[41] from PCR fragments or synthesized dsDNA (Integrated DNA Technologies) and verified by Sanger sequencing. For example, plasmid pCA3 was generated from plasmid pTrc99A-*mCherry-cI78*[EP] by replacing mCherry with PAmCherry. The PAmCherry fragment was generated using primers YH1 F and YH1 R. The plasmid pTrc99A-*mCherry-cI78*[EP8] was amplified using YH2 F and YH2 R to generate the second fragment. The two fragments were then added to a Gibson assembly reaction to enzymatically join the overlapping DNA sequences. Other plasmids were generated using similar methods with primers indicated in Supplementary Table 4. When relevant, homology regions for Gibson assembly are indicated in blue. Plasmids were introduced into their respective host strains by chemical transformation and selection for antibiotic resistance encoded by the plasmid. All plasmids are available on AddGene. The linker sequence used for PopTag[LL] was: DDAPAEPAAEAAPPPPPEPEPEPVSFDDEVLELTDPIAPE-PELPPLETVGDIDVYSPPEPESEPAYTPPPAAPVFDRD

### Growth conditions
Lysogeny broth (LB) and AB media were used as either a broth or solid for culturing bacteria. LB medium was used to grow E. coli BL21 Arctic Express (AE) and overnight cultures of E. coli MG1665. The minimal AB medium was used when inducing protein expression in E. coli MG1665 to ensure protein expression reproducibility, as all the components are defined; as opposed to a complex medium such as LB[28,42]. AB medium was supplemented with 0.2% of a carbon source (glycerol for growth or glucose to also inhibit basal protein expression from the $P_{trc}$ promoter), 0.2% casamino acids, 10 µg/ml thiamine, and 25 µg/ml uracil[28]. E. coli was grown in a 15 ml tube overnight for -15 h in 5 ml of LB broth at 37 °C on an orbital shaker at 200–225 rpm. Exponential phase cultures were prepared by diluting overnight cultures 1:100 and further incubated until an $OD_{600}$ of 0.2–0.6 was reached. When appropriate, the following chemicals were added to the medium at the indicated final concentrations: carbenicillin (100 µg/ml) for selecting the plasmids in culture, IPTG (0.1–1 mM) or L-arabinose (0.2%) for protein induction. Reagents used are listed in Supplementary Table 5.

### Total protein and immunoblot analyses
A 0.4-ml aliquot of E. coli cells ($OD_{600}$: 0.2–0.4) was lysed using a Qsonica cupped-horn sonication system (20 cycles, 30 s on, 10 s off at 30% power) and centrifuged at 10,000 x g for 1 min at 4 °C. The protein content in the supernatant was measured using a Bradford assay kit according to the manufacturer's instructions. To prepare samples for immunoblot analysis, an equal volume of 2 x Laemmli sample buffer was added to E. coli culture prior to boiling 20 min. One microgram of total protein from each sample was loaded in each lane of a 4-12% Bis-Tris NuPAGE gel. Gels were transferred onto a mini-size polyvinylidene difluoride membrane using a Trans-Blot Turbo system (Supplementary Table 5). The membrane was immunoprobed using a rabbit polyclonal antiserum against mCherry gifted by the Ming Li Lab at the University of Michigan (1:2000). The membrane was then incubated with the goat

anti-rabbit IgG Secondary Antibody IRDye 800CW. Membrane signals were visualized and quantified using LI-COR Image Studio. The mCherry band of each lane was normalized to the total intensity of the lane to calculate the cleavage level of mCherry fusion proteins.

## Wide-field fluorescence and phase-contrast imaging

Wide-field fluorescence and phase-contrast imaging were performed using a Nikon Ti2-E motorized inverted microscope controlled by the NIS Elements software with a SOLA 365 LED light source, a 100 × objective lens (Oil CFI Plan Apochromat DM Lambda Series for Phase Contrast), and a Photometrics Prime 95B back-illuminated sCMOS camera or Hamamatsu Orca-Flash 4.0 LTS camera. Fusions to mNeonGreen (mNG) were imaged using a "YFP" filter set (C-FL YFP, Hard Coat, High Signal-to-Noise, Zero Shift, Excitation: $500 \pm 10$ nm, Emission: $535 \pm 15$ nm, Dichroic Mirror: 515 nm). Fusions to msfGFP were imaged using a "GFP" filter set (C-FL GFP, Hard Coat, High Signal-to-Noise, Zero Shift, Excitation: $436 \pm 10$ nm, Emission: $480 \pm 20$ nm, Dichroic Mirror: 455 nm). DAPI fluorescence was imaged using a "DAPI" filter set (C-FL DAPI, Hard Coat, High Signal-to-Noise, Zero Shift, Excitation: $350 \pm 25$ nm, Emission: $460 \pm 25$ nm, Dichroic Mirror: 400 nm). mCherry was imaged using a "Texas Red" filter set (C-FL Texas Red, Hard Coat, High Signal-to-Noise, Zero Shift, Excitation: $560 \pm 20$ nm, Emission: $630 \pm 37.5$ nm; Dichroic Mirror: 585 nm).

## Time-lapse videos of protein induction in *E. coli BL21*

Gene *mNG-McdB* and *mNG-cI$^{ugg}$* were cloned into the multiple cloning site of the vector pET11a to create the pJB37 and pYH73 plasmids respectively, used for inducible expression under the control of a bacteriophage T7 promoter. pJB37 and pYH73 were transformed into BL21 (AE) cells and a 5 ml overnight culture containing 100 µg/ml of carbenicillin in LB medium was grown at 37 °C with shaking at 225 rpm. The overnight culture (50 µl) was used to inoculate 5 ml of LB supplemented with 100 µg/ml of carbenicillin in a 15 ml tube. The cells were grown at 37 °C with shaking at 225 rpm to an OD$_{600}$ of 0.2–0.6. Protein expression was then induced by the addition of 1 mM IPTG and 0.2% arabinose solution to the tube. Cells (2 µl) were spotted on a 1 cm diameter pad made of 1.5% UltraPure agarose in LB and supplemented with 1 mM IPTG and 0.2% arabinose. After 2 min, the cell-containing side of the pad was flipped onto a 35 mm glass-bottom dish and mounted onto the stage of a Nikon Ti2-E motorized inverted microscope controlled by NIS Elements software with a SOLA 365 LED light source, a 100 x objective lens (Oil CFI Plan Apochromat DM Lambda Series for Phase Contrast), and a Photometrics Prime 95B back-illuminated sCMOS camera. mNG was imaged using the "YFP" filter set (see above). An image series was captured every 1 min for 4 h.

## DAPI staining

*E. coli* BL21 (AE) cells with plasmid pJB37 were induced with 1 mM IPTG and 0.2% arabinose as described in the "Time-lapse videos of protein induction in *E. coli* BL21" section. DAPI (Supplementary Table 5) was added to the exponentially growing culture at a final concentration of 2 µM. Cells were incubated in DAPI for 15 min at 25 °C before imaging, without rinsing. DAPI was imaged using the "DAPI" filter set (see above).

## Localized cell lysis

pJB37 was transformed into BL21 (AE) cells, and a 10 ml overnight culture containing 100 µg/ml of carbenicillin was grown at 20 °C with shaking at 225 rpm. The overnight culture (0.5 ml) was used to inoculate 50 ml of LB supplemented with 100 µg/ml of carbenicillin in a 250 ml baffled flask. The cells were grown at 37 °C with shaking at 225 rpm to an OD$_{600}$ of 0.5. The flasks were plunged into an ice bath for 2 min. Protein expression was then induced by the addition of 1 mM

IPTG and 0.2% arabinose solution to the flask. Flasks were then returned to the shaker with incubation temperature set to 16 °C. Cells were then grown overnight with shaking at 225 rpm (~16 h induction). Cells (2 µl) were spotted on a 1 cm diameter pad made of 1.5% UltraPure agarose in LB. After 2 min, the cell-containing side of the pad was flipped onto a 35 mm glass-bottom dish and mounted onto the stage of a Nikon Ti2-E motorized inverted microscope controlled by NIS Elements software with a SOLA 365 LED light source, a 100 x objective lens (Oil CFI Plan Apochromat DM Lambda Series for Phase Contrast), and a Photometrics Prime 95B back-illuminated sCMOS camera. mNG-McdB was imaged using the "YFP" filter set (see above). A region of interest (0.5 µm diameter) was drawn at one cell pole and used as the target for a 405 nm laser pulse (1 sec at 50 mW), which caused localized cell lysis. An image series was captured every 500 ms for 60 s.

## Cellular shape change in *E. coli* BL21

*E. coli* BL21 (AE) cells with plasmid pJB37 and pYH73 were induced with 1 mM IPTG and 0.2% arabinose as described in the "Time-lapse videos of protein induction in *E. coli* BL21" section. The MreB inhibitor A22 (10 µg/ml, Supplementary Table 5) was added to the exponential phase cultures at time $t = 0$. The cultures were then incubated at 37 °C with shaking at 225 rpm for 6 h. Images were taken at 6 h post-treatment with the "YFP" filter set (see the "Wide-field fluorescence and phase-contrast imaging" section).

## Single-cell tunable induction of mCherry fusion focus

Gene *mCherry-cI$^{ugg}$, mCherry-PopTag$^{SL}$, mCherry-PopTag$^{LL}$, mCherry-mcdB, mCherry-mcdB$^{sol}$*, and *mCherry* were cloned into the multiple cloning site of the vector pTrc99A to create the pTrc99A-*mCherry-cI78$^{EP8}$*, pYH75, pYH80, pYH71, pYH86, and pYH77 plasmids respectively. The above plasmids were transformed into MG1665 cells and a 5 ml overnight culture containing 100 µg/ml of carbenicillin in AB medium (with 0.2% glycerol as the carbon source) was grown at 37 °C with shaking at 200 rpm. The overnight culture (50 µl) was used to inoculate 5 ml of AB (with 0.2% glycerol as the carbon source) supplemented with 100 µg/ml of carbenicillin in a 15 ml tube. The cells were grown at 37 °C with shaking at 200 rpm to an OD$_{600}$ of 0.2–0.6. 500 µM of IPTG was added to the exponential phase culture to induce fluorescent fusion protein expression. After each hour, 2 µL of the MG1665 culture were spotted on an agarose round pad. The pads were prepared by dissolving Ultrapure agarose in AB to a final concentration of 1.5%. A series of images were taken with the "Texas Red" filter set (see the "Wide-field fluorescence and phase-contrast imaging" section) every 1 h for 5 h.

## Nucleoid compaction

*E. coli MG1665* with pYH86 plasmid was induced with 1 mM IPTG for 2 h. Cells were then stained with 2 µM DAPI and spotted on an agarose pad containing 50 µM of ciprofloxacin (Supplementary Table 5). A series of images were taken with the "Texas Red" and "DAPI" filter sets (see the "Wide-field fluorescence and phase-contrast imaging" section) every 15 min for 7 h.

## Focus reversibility in response to changes in concentration in *E. coli* MG1665

*E. coli MG1665* with pTrc99A-*mCherry-cI78$^{EP8}$*, pYH75, pYH80, pYH71, pYH86, and pYH77 plasmids were induced with 1 mM IPTG for 2 h. The cells were then washed three times with 10 X volume of fresh AB media supplemented with 0.2% glucose and incubated for 30 min at 25 °C before imaging. After 30 min, 2 µL of the MG1665 culture was spotted on an agarose round pad. The pads were prepared by dissolving Ultrapure agarose in AB to a final concentration of 1.5% (the AB medium contains 0.2% glucose and 10 µg/ml cephalexin when specified). A series of images were taken with the "Texas Red" filter set (see the

"Wide-field fluorescence and phase-contrast imaging" section) every 15 min for 15 h.

## Osmotic Shock

*E. coli* MG1665 cells with pTrc99A-mCherry-cI78$^{EP8}$ and pYH75 plasmids were induced with 50 μM IPTG for 1 h. *E. coli* MG1665 cells with pYH80 and pYH71 plasmids were induced with 200 μM IPTG for 2 h. After a significant fraction of cells (> 20%) displayed foci, samples were washed three times with a 10 × volume of fresh AB media supplemented with 0.2% glucose and 300 mM KCl and incubated for 15 min at 37 °C with shaking. Control cells were also washed three times with a 10 × volume of fresh AB media supplemented with 0.2% glucose, but without KCl, and incubated at 37 °C with shaking. Cells (2 μl) were then spotted on a 1 cm diameter agarose pad. After 2 min, the cell-containing side of the pad was flipped onto a 35 mm glass-bottom dish and mounted onto a stage-top incubator. Images were captured with the "Texas Red" filter set (see the "Wide-field fluorescence and phase-contrast imaging" section). Cells treated with KCl were then subjected to another three washes with a 10 × volume of fresh AB media supplemented with 0.2% glucose, but without KCl, to visualize the focus dissolution. These cells, along with control cells that received no KCl treatment, were imaged as described above at 15 and 30 min following the osmotic upshift.

## Temperature shift

*E. coli* MG1665 with pTrc99A-*mCherry-cI78$^{EP8}$*, pYH75, pYH80, pYH71, pYH86, and pYH77 plasmids were induced with 200 μM IPTG for 2 h. The cells were then washed three times with 10 × volume of fresh AB media supplemented with 0.2% glucose and incubated for 1 h on ice. Cells (2 μl) were then immediately spotted on a 1 cm diameter agarose pad. After 2 min the cell-containing side of the pad was flipped onto a 35 mm glass-bottom dish and mounted onto a stage-top incubator with temperature control. The stage-top temperature was set up at 37 °C. An image series was captured with the "Texas Red" filter set (see the "Wide-field fluorescence and phase-contrast imaging" section) every 1 min for 30 min to observe focus dissolution.

## Fluorescence recovery after photobleaching (FRAP)

*E. coli MG1665* with pTrc99A-*mCherry-cI78$^{EP8}$*, pYH75, pYH80, and pYH71 plasmids were induced with 1 mM IPTG for 2 h. FRAP was performed after cells were washed with fresh AB media supplemented with 0.2% glucose and incubated for 30 min. A Nikon Ti2-E motorized inverted microscope controlled by NIS Elements software with a SOLA 365 LED light source, a 100 x objective lens (Oil CFI Plan Apochromat DM Lambda Series for Phase Contrast), and a Photometrics Prime 95B back-illuminated sCMOS camera were used to do FRAP experiments. Control images were taken before bleaching, then the regions of interest (foci) were bleached with a laser at 405 nm and 30% power (15 mW). A series of images was captured every 500 ms for 60 s after bleaching. More than ten different regions of interest (ROIs) were chosen per sample. Image analysis was performed using ImageJ. The background signal was subtracted from the bleached focus and an unbleached focus within the cell. The resulting focus signal was normalized such that the pre-bleach signal is one and the first frame post-bleaching is zero.

## Imaging the IbpA protein with wide-field fluorescence microscopy

The plasmids pTrc99A-*mCherry-cI78$^{EP8}$*, pYH75, pYH80, pYH71, and pYH77 plasmids were transformed into *E. coli* MG1665 that expresses IbpA-msfGFP by its native promoter. A 5 ml overnight culture containing 100 μg/ml of carbenicillin in AB medium (with 0.2% glycerol as the carbon source) of each strain was grown at 37 °C with shaking at 200 rpm. The overnight culture (50 μl) was used to inoculate 5 ml of AB (with 0.2% glycerol as the carbon source) supplemented with 100 μg/ml of carbenicillin in a 15 ml tube. The cells were grown at 37 °C with shaking at 200 rpm to an OD$_{600}$ of 0.2–0.6. 1 mM of IPTG was added to the exponential phase culture to induce fluorescent fusion protein expression for 2 h. 2 μL of the MG1665 cultures were spotted on an agarose round pad. The pads were prepared by dissolving Ultrapure agarose in AB to a final concentration of 1.5%. Images were taken with the "GFP" and "Texas Red" filter set (see the "Wide-field fluorescence and phase-contrast imaging" section) at 2 h post-induction.

## Imaging the IbpA protein with 2D Structured illumination microscopy (SIM)

Agarose pads with induced MG1665 cells were prepared as in the "Imaging the IbpA protein with wide-field fluorescence microscopy" section. SIM images were acquired on a Nikon N-SIM system equipped with a Nikon SR HP Apo TIRF 100× 1.49NA objective, a Hamamatsu ORCA-Flash4.0 camera (65 nm per pixel), and 488 nm and 561 nm lasers from a Nikon LU-NV laser launch. Cells were identified using DIC to avoid photobleaching. For each 2D-SIM image, nine images were acquired in different phases via the built-in 2D SIM modes. Super-resolution image reconstruction was performed using the Nikon Elements SIM module.

## Single-molecule fluorescence microscopy

Cells expressing protein fusions to PAmCherry of a plasmid with the P$_{trc}$ inducible promoter were grown in 3 ml of LB medium in a culture tube for ~ 14 h, with shaking at 225 rpm at 37 °C. The following day, cells were diluted 1:100 into 3 ml of fresh AB medium supplemented with 0.2% glucose or glycerol and grown to OD$_{600}$ ~ 0.3 before imaging either as is ("no induction" condition) or inducing with 100 μM IPTG ("inducing" condition). After a 2 or 4-h induction, cells were washed three times with 1 ml of AB medium supplemented with 0.2% glucose. All "no induction" and "inducing" cells were resuspended in M9 medium supplemented with 0.2% glucose for imaging. Agarose pads were made at 2% (w/v) with M9 minimal medium supplemented with 0.2% glucose. An aliquot of 2.5 μl of cells was loaded onto an agarose pad and sandwiched between two coverslips. Cells were imaged at room temperature with a 100 × 1.40 numerical aperture oil-immersion objective. A 406 nm laser (Coherent Cube 405–100; 0.2 W/cm$^2$) was used for PAmCherry photoactivation and a 561 nm laser (Coherent-Sapphire 561–50: 88.4 W/cm$^2$) was used for imaging. Given the difference in protein expression level, 200–400 ms activation doses were used for cells not induced and 50 ms activation doses were used for the 2-h induction sample. The 4-h sample had many preactivated molecules and molecules activated spontaneously without activation via 405 nm laser, which limited our ability to confidently localize single molecules. To reduce the number of molecules per imaging frame, an initial 405 nm laser activation (10–15 s) was followed by photobleaching through illumination with a 561 nm laser with the same power density as above for 15–20 min or until spatially resolved single molecules were observed. The fluorescence emission was filtered to eliminate the 561 nm excitation source and imaged at a rate of 40 ms/frame using a 512 × 512-pixel Photometric Evolve EMCCD camera.

## Single-molecule data analysis

Phase-contrast microscopy images were used to segment bacterial cells (see Image analysis for details) prior to localization and tracking. Single molecules were detected and localized with a 2D Gaussian fitting by the SMALL-LABS algorithm[43] and connected into trajectories using the Hungarian algorithm[44]. Localization heat maps (Fig. 2f) were made by normalizing the segmented cells and rotating them onto their long axes, followed by projection, binning, and

symmetrization of the single-molecule localizations onto the normalized cell[45,46].

Only single-molecule trajectories with a minimum of six steps were used for trajectory analysis. To extract the apparent diffusion coefficient for each trajectory, a modified version of the diffusion model[47] was fit to the mean square displacement (MSD) as a function of time lag, $\tau$, over the time interval $40 \leq \tau \leq 200$ ms. The modified diffusion model was used to account for motion blur due to averaging the true position of a molecule over the time of a single acquisition frame[47].

$$MSD = (8/3)D_{app}\tau + 4\sigma^2 \qquad (1)$$

In this case, $D_{app}$ is the apparent diffusion coefficient, $\tau$ is the time lag, and $\sigma$ describes the localization precision. Only fits with $R^2 \geq 0.7$ were kept. The log distribution histograms of the $D_{app}$ were each fit to a two-state Gaussian mixture to determine the $D_{app}$ and the associated weight fraction for the slow and fast diffusion modes (Supplementary Fig. 8).

### Measuring single-cell protein concentrations

To determine the average number of photons detected from a single mCherry molecule per imaging frame, cells expressing the indicated mCherry fusion proteins were grown in 3 ml of AB medium supplemented with 0.2% glucose to OD ~ 0.3 and washed once in 1 ml of M9 minimal medium prior to imaging. Cells were pre-bleached with a 561-nm laser until only a few isolated molecules were observed. Images were then recorded at 40 ms exposure with a 561 nm laser (110.5 W/cm²) and an input EM gain of 600 (NIS-Elements software setting). Single molecules were detected as described earlier. The number of photons detected per single molecule per imaging frame was obtained by calculating the integrated intensity counts of the 2D Gaussian fit from the localization step[48], which was then converted into the number of photons using the following camera calibrations.

The conversion gain (number of photoelectrons per fluorescence intensity count) calibration was performed as described in a Teledyne Photometrics Technical Note[49]. Briefly, multiple images of a white business card were acquired for 10-, 20-, 40-, 80-, 160-, and 320 ms exposure times with no electron multiplication (EM) gain. To account for the camera bias, 100 frames were acquired with nominally 0 ms exposure and a shuttered camera path. The average of these 100 frames was subtracted from every subsequently analyzed image. A plot of the mean signal of the image for each exposure time versus the variance of the same image gives a straight line with a slope that equals the conversion gain (Supplementary Fig. 4a). Pixel-to-pixel nonuniformity effects were removed by recording two images at each exposure time[50]. The conversion gain of our camera was found to be 1.40 electrons per intensity count.

To determine the EM gain (the number of electrons per photoelectron), two images were recorded: one long-exposure image (1 s) with no EM gain and one short-exposure image (10 ms) with an arbitrary EM gain. The bias was subtracted from both images. The EM gain multiplication factor is the factor difference in signal per time unit between the corrected images. We repeated this procedure for various software setting EM gains and found a linear fit in the range of 5–600 input EM gain with a conversion factor of 0.15 between nominal EM gain and output EM gain (Supplementary Fig. 4b).

To calculate the number of photons from the fluorescence image, we therefore multiplied the intensity counts recorded by the conversion gain of 1.40 and divided this quantity by the input EM gain multiplied by the EM gain conversion factor of 0.15. The distribution of photons per molecule per imaging frame was fit to a Gamma distribution resulting in a peak of ~90 photons per molecule per imaging frame (Supplementary Fig. 4c).

To determine the apparent cellular concentration of mCherry-McdB and mCherry-PopTag^LL when foci are present, cells expressing these proteins were grown and prepared as described above. The mCherry-McdB strain was induced with 1 mM IPTG for 2 h prior to imaging and the mCherry-PopTag^LL was induced with 100 μM IPTG for 2 h prior to imaging. Cells were imaged with a 561 nm laser (110.5 W/cm²) at 20 ms exposure and 10x (NIS-Elements software setting) EM gain. To minimize the effect of photobleaching on the photon counting measurement, image acquisition was started prior to laser illumination.

The brightest five frames in the movie were averaged to determine the cell brightness before photobleaching. The integrated total fluorescence emission within a cell is linearly related to the total quantity of McdB or PopTag^LL molecules per cell: this intensity value was divided by the number of photons per mCherry molecule per imaging frame to obtain the McdB or PopTag^LL copy number per cell. To determine the cellular concentration of the proteins, the volume of the cell was estimated by modeling it as a cylinder with spherical caps[51].

### Image analysis

Cell segmentation was performed with the Cellpose[52] and Omnipose[53] packages in Python. To train the Cellpose model optimally for bacterial cell morphology, 26 raw phase contrast images of cells were manually annotated. To segment the cells using the trained model, a Gaussian blur (standard deviation of Gaussian = 0.066 μm) was applied to the bacterial cell phase-contrast images, and the blurred cells were segmented. Cells touching the borders of the image were ignored. Erroneous segmentations were manually corrected using the Cellpose GUI or excluded from further analysis.

To calculate the condensation coefficient of a cell, the fluorescence intensity in each pixel in a cell was corrected for background by subtracting the median value of all pixels in an image outside of the cell regions and normalized by the minimum and maximum pixel intensity values in the corresponding cell:

$$I_n = (I - I_{min})/(I_{max} - I_{min}) \qquad (2)$$

These normalized pixel intensities were then binned to generate histograms that represent the localization pattern for each protein[16,29,54] (Supplementary Fig. 4a–f).

Next, cells were classified by the presence or lack of a focus. An ROI was defined for each cell using the segmentation phase mask. Each ROI was applied a Gaussian blur (SD = 0.066 μm) and normalized using Eq. (2). Next, foci were detected by generating a binary image where only pixels above a specified intensity threshold were assigned a non-zero value. The putative foci were then filtered by intensity, area, and eccentricity (see Supplementary Table 6 for parameters used). Homogeneously distributed protein mCherry or cells without a detected focus display a flat distribution whereas the strongly clustered protein cI^agg displays a strongly left-skewed distribution (Supplementary Fig. 5a–f). To quantify the differences between the proteins measured, we calculated the fraction of pixels with a normalized intensity below a threshold value, $I < 0.3$, 0.5, and 0.7, for each cell (Supplementary Fig. 5g). These values were then normalized to the condensation coefficient distribution of mCherry as our experimental representative of a homogenous distribution (Fig. 3b). Condensation coefficients calculated for cells in the A22 treatment (Supplementary Fig. 1c-d), localized lysis (Supplementary Fig. 1e-f), and temperature shift (Fig. 4k-l) experiments were not normalized to a homogenous mCherry-only sample.

Foci partition ratios were calculated from cells with a detected focus as the ratio of the average background-corrected pixel intensity

within a focus to the average background-corrected pixel intensity of the cells excluding the focus region. For cells with two or more detected foci, the average intensity of all foci was used for the ratio. Fluorescent foci were tracked during the reversibility experiment for images acquired over the course of bacterial cell division events or during cell elongation with cephalexin treatment as described above. Because the increased cell density over the imaging period causes cells to move out of focus and prevents accurate quantification of the data, we stopped our analysis at 10 and 7.25 hrs, for the generational dilution and cephalexin treatment, respectively. We detected fluorescent foci by the LoG method in TrackMate[55] using an estimated blob diameter of 600 nm and a quality threshold of 80 for cI[agg], PopTag[SL], PopTag[LL], and 225 for McdB. Spot detections were subsequently linked to generate trajectories using the Simple Lap Track option with a maximum linking distance of three pixels (198 nm), a maximum gap-closing distance of 3 pixels (198 nm), and a maximum gap of 2 frames (30 min). The following trajectory filters were used: (i) only trajectories present in the first frame were tracked in subsequent frames, (ii) trajectories of foci in cells that move out of the field of view during the movie were excluded, (iii) false positive detections in the first frame were removed, and (iv) trajectories of foci in cells that did not grow (cephalexin treatment) or divide (generational dilution) were excluded. Erroneous trajectories were removed from further analysis. Cells that exhibited focus dissolution over the imaging period were then segmented at the frames adjacent to the dissolution event. We then measured the mean fluorescence intensity of the cell in both frames (Figs. 4c and f). Focus lifespans were calculated for foci that dissolved prior to the end of the imaging period. See Supplementary Table 1 for statistical details.

For cells that reformed a fluorescent focus after cell division, we segmented the mother cell in the preceding frame and the two daughter cells. The total fluorescence intensity of each daughter cell was then normalized to that of the mother cell (Fig. 4h–j).

Images of dual-color labeled cells expressing msGFP-IbpA and mCherry fusion proteins were analyzed by detecting fluorescent foci in the mCherry channel using an intensity-based threshold of 30% of the maximum intensity in the image. mCherry fusion protein spot detections were subsequently filtered based on a minimum area threshold of 4 pixels and an eccentricity threshold of 0.75. The centroid of these spots was used to define the center of a $23 \times 23$ pixel ROI. This ROI was then used to analyze the IbpA channel. First, an intensity profile was collected using the centroid of the ROI as the center point for both channels and subsequently normalized to the max intensity values in the mCherry channel. Next, the normalized ROIs for each channel were averaged to generate the projection images (Fig. 6c). The resulting intensity patterns for both channels were then fit with a 2D Gaussian to measure the full width at half max (FWHM) of the intensity. Finally, we calculated the ratio of $FWHM_{IbpA}$/$FWHM_{mCherry}$.

### Reporting summary
Further information on research design is available in the Nature Portfolio Reporting Summary linked to this article.

## Data availability
All data generated or analyzed during this study are included in this published article and its supplementary information files. Source data are provided with this paper.

## Code availability
Data processing and analysis scripts for this study were written in MATLAB and Python. The code generated for this study is available on Github: All image analysis scripts (https://github.com/BiteenMatlab/bacterial_condensates)[56], SMALL-LABS algorithm package (https://github.com/BiteenMatlab/SMALL-LABS).

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

## Acknowledgements

This work was funded by NSF CAREER award 1941966 to AGV and NIH award R01GM144731 to JSB. We would like to thank Dr. Sarah Veatch for providing critical feedback on experimental direction and data interpretation. We also thank Dr. Eric Rentchler of the Biomedical Microscopy Core at the University of Michigan for training and coordinating access to 2D-SIM. We thank Dr. Ming Li's Lab at the University of Michigan for the mCherry antibody, Dr. Abram Aertsen's Lab for sending plasmids and *E. coli* strains. We thank Dr. Julien Mortier for constructing the strain that expressed IbpA-msfGFP with its native promoter in *E. coli*.

## Author contributions

Conceptualization, Y.H., C.A.A., J.S.B., and A.G.V.; Methodology, Y.H., C.A.A, R.E.D., and M.G.; Fomal Analysis, Y.H., C.A.A., J.S.B., and A.G.V.; Investigation, Y.H., C.A.A.; Resources, J.S.B. and A.G.V.; Writing—Original Draft, Y.H., C.A.A., J.S.B., and A.G.V.; Visualization, Y.H., C.A.A, R.E.D. and M.G.; Supervision, Y.H., C.A.A., J.S.B., and A.G.V.; Funding Acquisition, J.S.B. and A.G.V.

## Competing interests

The authors declare no competing interests.

## Additional information

**Supplementary information** The online version contains
supplementary material available at

Julie S. Biteen or Anthony G. Vecchiarelli.

**Peer review information** *Nature Communications* thanks Wei Zhao, and
the other, anonymous, reviewer(s) for their contribution to the peer
review of this work. A peer review file is available.

