## [Peer Review File · Nature Communications]

Reviewers' Comments:

Reviewer #1:

Remarks to the Author:

The study by Hoang et al. sets out to develop an experimental framework in *E. coli* to be able to quantitatively characterize and importantly compare bacterial condensates from diverse proteins and species in a standardized system with controls. With the exception of the IbpA reporter, which is novel and intriguing, the other approaches are not individually novel. But collectively the experiments do make headway in establishing a set of standardized measurements and controls in *E. coli* to characterize phase separation and the biophysical properties of bacterial condensates. This is an ambitious goal for the field, but in order to be a generally useful set of quantitative tools, the authors need to do additional analysis of their existing data, and also perform several additional controls, as outlined in 'Major Comments' below.

Major Comments:

1. Differences between mNG-McdB in *E. coli* BL21 and mCherry-McdB in MG1655. The authors should explain why they switched tags – mNG is brighter, and mCherry is known to drive artifactual foci formation (Landgraf et al., *Nat Methods* 2012, PMID: 22484850), so why mCherry? Also, the distinction the authors draw between 'overexpression' and 'heterologous expression' is not clear/accurate. I think it would be more accurate to say 'switch-like' expression and 'single-cell tunable expression' to describe their two expression systems. Also, the cells are clearly sick in their 'overexpression' experiments, and they rightly point out that the properties of the condensates might be very different under these conditions than at lower concentrations. But several of their key claims (fusion, dissolution upon temperature shift and volume change) come from these 'overexpression' experiments. Therefore it would be important to repeat a few of the key experiments from Fig S1 using the mCherry-McdB/MG1655 system, most notably the temperature dependence, which is the most compelling of this set. Finally and most importantly, the authors should also compare these measurements (or at least the temperature shift) with the other model constructs (PopTags and cIagg). Such comparisons with the 'control' proteins are the real strength of their approach.

Other specific concerns with the mNG-McdB data and conclusions drawn:

- i. Fig S1B: claim that fusion events are observed is not supported by these data – the foci approach each other, but that's not enough to firmly establish fusion events. Either this claim should be removed, or more work should be done to quantify fusion events (not trivial).
- ii. Fig S1D: why are the condensates so much smaller and differentially localized in this experiment than the rest of the paper? Do temperature shifts not dissolve larger polar condensates? If they don't, the authors should report this too.
- iii. Fig S1E: it is not clear from these results that increasing the cell volume dissolves condensates, particularly because there are phase bright foci visible in the phase contrast image which co-localize with the mNG fluorescence that are visible even in the last frame of the time series. Also, the fluorescence does not become uniform in these cells even after 12h, and the total fluorescence intensity seems to have increased and yet is not quantified.
- iv. Fig S1F: Polar lysis experiments are difficult to interpret without a comparison to something that does not become solubilized, like the cIagg control.

2. Show cytoplasmic concentration as the independent variable, not just time. My biggest general concern with the paper: after doing all the careful controls, particularly Fig S4, in order to be able to convert fluorescence signal into cellular protein concentration, the authors then do not fully exploit this approach by applying it to the majority of their measurements. They plot time as the independent variable, when in fact cytoplasmic concentration is in many cases more powerful independent variable. For example, in Figure 2B: time post induction vs. % cells with a fluorescent focus is important, but the complementary and key plot to show in parallel would be concentration vs. % cells with a fluorescent focus across all timepoints. If the expression system they are using is indeed tunable at the single cell level, then quantifying the data they already have collected in this manner would allow them to sample a much broader range of concentrations. By picking only the 4h timepoint in Fig 2c, they can only sample a very narrow range of concentrations (a 3-fold range from ~50 – 150 μM for McdB), which weakens their

apparent C_{sat} estimates. It would also be important to test/show if focus formation is purely concentration dependent, and independent of induction time. If there is a time dependence that can't be explained by concentration, this is also useful to know and may lead to more information about the properties of the condensates, such as aging.

The other place where the concentration dependence needs to be shown is in the dissolution during outgrowth experiments in Figure 3. The focus intensity as a function of time shown in Fig 3b is a bit confusing and suggests that the dissolution of these foci is not, in fact, switch-like once concentrations drop below C_{sat} . Plotting the concentration at which a focus dissolves might be a more revealing metric. Also, the focus-reformation observation after cell division is very interesting, but the proposed explanation does not make sense unless there is an asymmetric distribution of the protein between daughter cells after cell division. This model can and should be directly tested by measuring the total intensity/concentration in each daughter cell.

3. Chicken or egg problem with IbpA foci. IbpA-msfGFP forms foci at the poles in the absence of heterologous expression of any of these proteins, as clearly shown in figure S10A. Is IbpA, a native *E. coli* protein, partitioning into these heterologous condensates, or rather are PopTag and McdB proteins being recruited to inclusion bodies and changing their properties? Either result is interesting. It is also possible that these heterologous condensates form a second phase around existing inclusion bodies (SIM data suggests there may be two phases?). There is a lot of interesting work to do with this that is beyond the scope of this study, but one key control for this study that can and should be done with the strains in hand is to show timelapse movies of the various mCherry constructs being induced in the IbpA-msfGFP strain. In this manner, the authors can demonstrate if pre-existing IbpA foci nucleate the heterologous condensates, or if IbpA is recruited away from putative inclusion bodies.

Additionally:

- i. Given that the condensates studied go to the poles, it is also worth pointing out that we know that the bacterial cytoplasm is inhomogeneous and that the cell pole have distinct biophysical properties, including attenuated diffusion (Smigiel WM et al., 2022, PMID: 35960807). This should be discussed.
- ii. The claim that IbpA can distinguish between condensates and aggregates may be an overstatement, given that the authors only look at a single aggregation-prone construct. Soften the language here, or include another aggregation-prone protein to get at generalizability. The best case would an aggregation-prone variant of PopZ or McdB.

More Minor Points:

Fig 2 panels B vs C: At 4h the percentage of cells with a focus in panel B is ~50% for McdB, but density of points in panel C is nowhere near 50% with foci, 50% without. Instead, 170 cells have a focus, and 834 cells do not. If the ratio is actually closer to 50/50, why were only 1/4 as many cells with foci detected/quantified?

E. coli as a model system: the authors should more clearly motivate using *E. coli* as the model 'test tube' for expressing heterologous proteins from species separated by hundreds of millions of years of evolution, and characterizing the resulting condensates. I don't disagree with this choice, but it is important to explicitly define why it is informative to use *E. coli* instead of a the protein's native organism, or a eukaryotic system like yeast, where the diffraction limit would be less of an issue.

Line 47: Introduce more about McdB: the specific organism it is from, which is not mentioned, its properties that make it an interesting condensate-forming protein to study, and in both the introduction/discussion, point out the significance of the measurements made in this manuscript to broader questions about the function of McdB. This connection is missing and important. The same could be said for the PopTag 'controls:' did we learn anything new that is functionally relevant for PopZ's function in *C. crescentus*? Or just confirm our existing understanding? Or raise new questions?

Line 179-180: Confusing wording: "expression was stopped to maintain a constant cellular protein level." It is not a constant level, the point is that removing the inducer but allowing growth to

continue means the protein level will drop, as is written in the next line.

Line 142 and discussion: A $csat_{app}$ in vivo is estimated, but the significance of the value is not discussed relative to what is known, and without this, it is not a particularly meaningful measurement. Some context that might make these measurements more meaningful would be: (1) comparison with the cellular concentration of PopZ and McdB in vivo in their native organisms (and the authors can presumably do this for McdB, since they are pioneering this system). If the apparent $csat$ from *E. coli* is quite different from this, it would be interesting and useful to speculate why – due to differences in the subcellular interaction partners in the native system? Due to other conditions in the native host environment? (2) comparison with in vitro measurements

Reviewer #2:

Remarks to the Author:

The understanding of biomolecular condensates in bacteria is physiologically important but limited in technology. In this study, an experimental framework was developed to assess whether a protein focus could be determined as a phase-separated condensate. The formation, reversibility, and dynamics of condensate-forming proteins were tested in *Escherichia coli* by $csat$ calculation, changes in cell shape or temperature, FRAP, and single-molecule tracking. The topic is no doubt interesting and important. However, there are still some major issues which need to be addressed. For example, some conclusions may not be supported by the results or being overstated in the manuscript. Some data interpretations of methods need to be cautious. Please see below for more details.

1. The results of Fig S1 are lacking of quantifications, including those of cell curvature localizations, temperature shifts and drug-induced changes of mNG-McdB.

“We speculate mNG-McdB foci wet to the membrane via nonspecific electrostatic associations and locally occlude cell wall synthesis.” Any evidence for this claim? What activity of McdB could cause the local changes in cell wall? Please clarify this.

“The resulting asymmetry in cell wall growth thereby induces cell curvature.” Any evidence for the asymmetry in cell wall growth?

2. As stated by the authors that the temperature shifts and drug-induced changes to cell volume may involve pleiotropic effects, the claim of “suggesting that the cellular levels of mNG-McdB dropped below $csat$ ” needs to change to “indicating that the cellular levels of mNG-McdB may drop below $csat$ ”.

Fig S1f, if it is possible that the shift of fluorescent signal to the opposite end is the result of protein de-novo synthesis, rather than the solubilization of the opposing focus? Please quantify the results and explain this.

3. What’s the difference between mCherry-mcdB (Fig 2) and mNG-McdB (Fig S1) constructions? Why the authors emphasize that mCherry-mcdB (Fig 2) is the tunable expression? The photoactivatable fluorescent proteins (PAmCherry fused proteins) in Fig 2 could be tunable constructions. However, only the turn on of expression but no turn off of expression was shown. Hence, no big difference was demonstrated in comparison to the IPTG induction.

4. Why to choose the five proteins of cIagg, PopTagSL, PopTagLL, McdB, and McdBsol to demonstrate the framework in this study? Are they being selected randomly? What’s the logic behind these five proteins? In addition, the authors should describe the PopTag more specifically, so does the linker for PopTagLL. Is PopTagLL using PopZ sequence as the protein linker?

5. The main purpose of western blot in Fig S2 should indicate the expression levels of mCherry fusion proteins, instead of the degradation levels of these proteins. Nevertheless, I don’t think it’s a good idea to use western blot to indicate the degradation levels due to the non-specificity of

antibody.

What I understand is that `csat_app` in Fig 2 is derived from Fig S4 quantification. However, what's the quantitative signal used in Fig S4 and how to transform this signal into protein concentration. Please clarify this.

6. "Intriguingly, PAmCherry-McdBsol also formed high-density regions at the poles, consistent with bulk fluorescence measurements (see Fig. 2a)." Why their localization patterns (in Fig. 2a and Fig. 2f) are different?

"we speculate that the localization pattern of the fluidized PAmCherry-McdB condensate is due to nucleoid exclusion by repulsive electrostatic interactions." Any evidence for this claim, just because the net negative charge of its IDR was increased? The mCherry alone could be a possible control in Fig 2f.

7. Fig S6, a quantification could be performed to support the conclusion here.

8. The interpretation of Fig 3 need to be more cautious because it has a very complicate environment in living cells. The authors did not take the protein degradation or expression leaky into consideration. Also, the authors need to be careful about the photobleaching during the time lapse imaging.

9. "As the cell divides, the volume decreases, leading to an increase in the concentration back above `csat` in the daughter cell, which ultimately results in the reformation of a focus." The interpretation for Fig S8 makes me confused, since the protein expression was stopped and the concentration will not be increased after cell division.

10. The results in fig 4 are more reliable. Could the authors provide more details to support the accessibility of this method in the framework? For example, how many molecules was used to probe in fig 4c? What is the variance distribution in this study?

11. It has been demonstrated that IbpA could serve as a molecular sensor for protein aggregates. However, the penetration of condensates by IbpA was not shown until this work. Since people use this method to discriminate the aggregates and functional proteins before, much more evidences should be provided to support this claim. Two proteins are not having statistical significances.

12. No text lines.

Point-by-point Response:

Reviewer #1 (Remarks to the Author):

The study by Hoang et al. sets out to develop an experimental framework in *E. coli* to be able to quantitatively characterize and importantly compare bacterial condensates from diverse proteins and species in a standardized system with controls. With the exception of the lbpA reporter, which is novel and intriguing, the other approaches are not individually novel. But collectively the experiments do make headway in establishing a set of standardized measurements and controls in *E. coli* to characterize phase separation and the biophysical properties of bacterial condensates. This is an ambitious goal for the field, but in order to be a generally useful set of quantitative tools, the authors need to do additional analysis of their existing data, and also perform several additional controls, as outlined in 'Major Comments' below.

Major Comments:

1. Differences between mNG-McdB in *E. coli* BL21 and mCherry-McdB in MG1655. The authors should explain why they switched tags – mNG is brighter, and mCherry is known to drive artifactual foci formation (Landgraf et al., Nat Methods 2012, PMID: 22484850), so why mCherry?

We fused our protein set to mCherry for the following reasons: (1) Ultimately, our studies in MG1655 began with our lbpA experiments, where lbpA is fused to sfGFP and expressed from the native locus. This construct was generated and kindly provided by the Aertsen lab (Govers et al., 2018). We therefore required another fluorescent protein for our protein set that can be co-imaged with lbpA-sfGFP, such as mCherry. (2) PAmCherry remains a preferred fluorescent protein fusion for single-particle studies, given its low propensity for blinking. mCherry and PAmCherry differ in sequence by only 10 amino acids, which allowed us to reliably compare our single-particle data using PAmCherry with our wide-field fluorescence microscopy experiments, using mCherry. We have added an explanation for the mNG to mCherry switch:

Line 136: We switched from mNG to mCherry for our experiments in MG1665 because we later performed single-particle tracking of our protein set using PAmCherry (see Fig. 3 and 5). Therefore using mCherry for our wide-field fluorescence microscopy, instead of mNG, allowed for a reliable comparison with our single particle data. In addition, later experiments required our protein set to be co-imaged with an E. coli chaperone protein fused to sfGFP (see Fig. 6).

We are aware of the Landgraf study showing that mCherry and other fluorescent proteins may drive artifactual focus formation of proteins with oligomeric potential. Indeed, the use of any fluorescent fusion has several other potential caveats that also require consideration. For example, mNG and other fluorescent fusions, have been shown to provide the opposite effect: solubilizing proteins and thereby changing c_{sat} relative to the unlabeled protein. For these reasons, in these assays, it is critical to ensure (1) the protein set is fused to the same fluorescent protein, regardless of the fusion type chosen, and (2) the comparisons of the protein set are relative in nature.

We note that mCherry-PopZ fully complements a PopZ deletion in *Caulobacter crescentus*, and with no notable differences in the PopZ micro domain formed at the cell pole as shown by fluorescence microscopy and CryoEM (Lasker et al, 2022). As for McdB, our McdB^{sol} mutant is defective in forming condensates *in vitro*, however it still forms a hexamer like wildtype McdB (Basalla et al., 2023). Our data here shows that hexameric McdB^{sol} remains fluidized *in vivo*, even when fused to mCherry.

Also, the distinction the authors draw between 'overexpression' and 'heterologous expression' is not clear/accurate. I think it would be more accurate to say **'switch-like' expression and 'single-cell tunable expression' to describe their two expression systems.**

As suggested by the reviewer we now use 'switch-like expression' and 'single-cell tunable expression' to describe the two expression systems throughout the paper. Thank you.

Also, the cells are clearly sick in their 'overexpression' experiments, and they rightly point out that the properties of the condensates might be very different under these conditions than at lower concentrations. But several of their key claims (fusion, dissolution upon temperature shift and volume change) come from these 'overexpression' experiments. Therefore it would be important to repeat a few of the key experiments from **Fig S1** using the mCherry-McdB/MG1655 system, most notably the temperature dependence, which is the most compelling of this set. Finally and most importantly, the authors should also compare these measurements (or at least the temperature shift) with the other model constructs (PopTags and clagg). Such comparisons with the 'control' proteins are the real strength of their approach.

We agree that the temperature dependence is one of the most compelling pieces of data, but not ideal, when performed in the overexpression experiments. As suggested by the reviewer, we now provide the temperature shift experiments for the protein set in MG1665. As expected, we observed a redistribution of fluorescence for McdB, PopTag^{SL}, and PopTag^{LL}, but not cl^{agg} (Fig. 4k and Supplementary Video 8). The condensation coefficients were also quantified and provided (Fig. 4l).

Other specific concerns with the mNG-McdB data and conclusions drawn:

i. **Fig S1B:** claim that fusion events are observed is not supported by these data – the foci approach each other, but that's not enough to firmly establish fusion events. Either this claim should be removed, or more work should be done to quantify fusion events (not trivial).

We agree that the resolution is too limited for us to firmly establish a liquid-like fusion event, as opposed to other possibilities. We have therefore removed this data and claim, as suggested by the reviewer (In 99-104).

ii. **Fig S1D:** why are the condensates so much smaller and differentially localized in this experiment than the rest of the paper? Do temperature shifts not dissolve larger polar condensates? If they don't, the authors should report this too.

In the experiment previously shown in Fig. S1D, mNG-McdB expression was slowly induced and with high illumination intensity. This allowed us to observe the initial formation of multiple foci. As soon as these foci were observed, the temperature was ramped up to 37 °C on the stage-top incubator and these multiple foci dissolved, despite the protein concentration continuing to increase. These multiple smaller foci eventually fused to form the large polar foci observed in all subsequent experiments in the paper. Without the temperature increase, and with faster induction conditions, mNG-McdB and all other control proteins were observed to form a single polar focus at the pole, without the display of smaller foci first. We now show that even these large polar foci of McdB dissolve upon temperature shift in MG1665 cells (Fig. 4k-l). Thank you for the suggested addition, which has significantly strengthened our conclusions.

iii. **Fig S1E:** it is not clear from these results that increasing the cell volume dissolves condensates, particularly because there are phase bright foci visible in the phase contrast image which co-localize with the mNG fluorescence that are visible even in the last frame of the time series. Also, the fluorescence does not become uniform in these cells even after 12h, and the total fluorescence intensity seems to have increased and yet is not quantified.

Indeed, the mNG-McdB foci in some cells have not fully dissolved by the end of Video 2. However, all foci across the entire cell population are dissolving into the cytoplasmic phase, despite the fact that protein concentration continually increased. We now add mNG-cl^{agg} to compare and quantify the stark differences in dissolution of the McdB focus after A22 treatment versus cl^{agg} foci remaining intact with no dissolution (Fig. S1c-d, Supplementary Video 3).

These experiments are difficult to perform in MG1665 cells because they divide quickly, and cell division convolutes whether the foci dissolve due to cell shape change or due to generational dilution of the protein as we provide evidence for later in the manuscript.

iv. **Fig S1F:** Polar lysis experiments are difficult to interpret without a comparison to something that does not become solubilized, like the clagg control.

We have now added the cl^{agg} control as suggested and provide quantification (Fig. S1e & f). As expected, localized cell lysis does not destabilize cl^{agg} foci. Thank you for the useful suggestion.

2. Show cytoplasmic concentration as the independent variable, not just time. My biggest general concern with the paper: after doing all the careful controls, particularly Fig S4, in order to be able to convert fluorescence signal into cellular protein concentration, the authors then do not fully exploit this approach by applying it to the majority of their measurements. They plot time as the independent variable, when in fact cytoplasmic concentration_{app} is in many cases more powerful independent variable. For example, in Figure 2B: time post induction vs. % cells with a fluorescent focus is important, but the complementary and key plot to show in parallel would be concentration_{app} vs. % cells with a fluorescent focus across all timepoints. If the expression system they are using is indeed tunable at the single cell level, then quantifying the data they already have collected in this manner would allow them to sample a much broader range of concentrations. By picking only the 4h timepoint in Fig 2c, they can only sample a very narrow range of concentrations (a 3-fold range from ~50 – 150 μM for McdB), which weakens their apparent Csat estimates. It would also be important to test/show if focus formation is purely concentration dependent, and independent of induction time. If there is a time dependence that can't be explained by concentration, this is also useful to know and may lead to more information about the properties of the condensates, such as aging.

The original measurements for the data presented in Fig. 2a-c were performed on an epifluorescence microscope with a sCMOS camera. Our initial intention was to perform the necessary calibrations to convert the fluorescence counts here into concentrations, however we were unable to see single molecules under LED illumination using this optical setup. Therefore, we performed the calibrations on a single-molecule fluorescence microscopy setup with an EMCCD, which uses a 561 nm continuous wave laser as the illumination source, and expressed mCherry-McdB or mCherry-PopTag^{LL} to levels where we would see cells with and without a focus (Fig. 2d).

We agree with the reviewer that plotting the apparent concentration versus cells with a fluorescent focus is a powerful depiction of the data. While with the data collected, we cannot calculate molar concentrations, we now plot the single-cell fluorescence concentration (sum of pixel intensities divided by cell area) against induction time. Across all time points, and to the reviewer's point, we can now show that cells with a fluorescent focus exhibit, on average, a larger fluorescence concentration than cells without a

fluorescent focus. Along these lines, we also now observe a discrete fluorescence concentration at which the McdB and PopTag^{LL} cells form a focus (Fig. 2c). These values compare well to the molar concentrations we determined on the other imaging system (Fig. 2d).

This re-analysis of the data inspired by these reviewer comments have significantly strengthened the interpretation and conclusions of the paper. Thank you.

The other place where the concentration dependence needs to be shown is in the dissolution during outgrowth experiments in Figure 3. The focus intensity as a function of time shown in Fig 3b is a bit confusing and suggests that the dissolution of these foci is not, in fact, switch-like once concentrations drop below C_{sat} . Plotting the concentration at which a focus dissolves might be a more revealing metric.

The reviewer makes a great point. We interpreted the decrease of foci intensity over time as a decrease in condensate size as the concentration decreases. Once the cellular concentration reaches the saturation concentration, the foci do dissolve in a 'switch-like' manner. We agree with the reviewer that plotting this concentration is an improved metric. Therefore, we removed the focus intensity as function of time panel and replaced it with single-cell fluorescence concentration at the point of focus dissolution (Fig. 4c, f). The values for PopTag^{LL} and McdB are consistent with the measurements in Fig. 2c. We thank the reviewer for this useful suggestion.

Also, the focus-reformation observation after cell division is very interesting, but the proposed explanation does not make sense unless there is an asymmetric distribution of the protein between daughter cells after cell division. This model can and should be directly tested by measuring the total intensity/concentration in each daughter cell.

We now quantified focus reformation after cell division by plotting the total fluorescence intensity of each daughter cell normalized to that of the mother cell and classified the daughter cells by the presence or absence of a focus. On average, the daughter cells with a focus have a higher fluorescence intensity level than those without a focus (Fig. 4h,j). The findings support the proposal that focus reformation is due to an asymmetric inheritance of fluorescence intensity/protein concentration after cell division.

3. Chicken or egg problem with IbpA foci. IbpA-msfGFP forms foci at the poles in the absence of heterologous expression of any of these proteins, as clearly shown in figure S10A. Is IbpA, a native *E. coli* protein, partitioning into these heterologous condensates, or rather are PopTag and McdB proteins being recruited to inclusion bodies and changing their properties? Either result is interesting. It is also possible that these heterologous condensates form a second phase around existing inclusion bodies (SIM data suggests there may be two phases?). There is a lot of interesting work to do with this that is beyond the scope of this study, but one key control for this study that can and should be done with the strains in hand is to show timelapse movies of the various mCherry constructs being induced in the IbpA-msfGFP strain. In this manner, the authors can demonstrate if pre-existing IbpA foci nucleate the heterologous condensates, or if IbpA is recruited away from putative inclusion bodies.

The conclusion from our findings is that IbpA associates differently with condensates than it does with solid aggregates, and as such, the nature of IbpA association, and potentially the nature of association with other chaperones, can be used as an *in vivo* marker for the material state of a focus *in vivo*. Determining the order and mode of association is indeed interesting, but falls outside the scope of this paper and is not necessary for justifying the conclusions currently provided in the manuscript. We do look forward to future studies focused on how chaperones associate with foci of varying material states in bacteria.

Although these experiments are not required to justify our conclusions currently in the paper, we did attempt the following experiments to potentially address how IbpA is recruited to these foci as requested by the reviewer. We induced expression of our protein set and observed focus formation, but we could not resolve whether these foci recruited IbpA or vice versa. The difficulty lies in the fact that IbpA-msfGFP is expressed from its native locus in our current strain, and it is always making several dynamic puncta of IbpA across the cell; some of which are very faint. When foci from our protein set first form, they are also very small. Therefore, we cannot discern if colocalization is because the foci of our protein set were there first, or if dynamic IbpA foci were there first. Live-cell microscopy with higher spatiotemporal resolution, combined with a controlled IbpA-GFP induction strain, is likely required to determine how IbpA is recruited.

Additionally:

i. Given that the condensates studied go to the poles, it is also worth pointing out that we know that the bacterial cytoplasm is inhomogeneous and that the cell pole have distinct biophysical properties, including attenuated diffusion (Smigiel WM et al., 2022, PMID: 35960807). This should be discussed.

We now mention the findings of this report. Specifically, the inhomogeneity of the bacterial cytoplasm and the unique biophysical environment of the cell poles, such as attenuated diffusion. We agree that it is important to note that all structures studied here are, not only fused with the same fluorescent protein, but all are located at the cell poles, as opposed to other regions of the cell with faster protein diffusion. This point further strengthens the validity of our comparisons. Thank you for the suggestion. We have added the following:

A recent study reports on the inhomogeneity of the bacterial cytoplasm. Specifically, slowed protein diffusion at the cell poles compared to other regions of the E. coli cell (Smigiel WM et al., 2022). The finding highlights the importance of comparative studies of in-focus diffusion measurements for foci occupying the same cellular location. It is likely that slowed protein diffusion contributes to condensate formation at cell poles. But even if diffusion is slowed at the poles by some degree, our in-focus diffusion measurements are much slower and non-uniform across our protein set; supporting the proposal that interactions involved in condensate formation further affect the diffusion of the proteins. Collectively, the data parse the dynamic exchange of molecules in foci and the surrounding cytoplasm, and delineate the spectrum of possible mobilities within a focus, thus allowing for inference of its material state.

ii. The claim that IbpA can distinguish between condensates and aggregates may be an overstatement, given that the authors only look at a single aggregation-prone construct. Soften the language here, or include another aggregation-prone protein to get at generalizability. The best case would an aggregation-prone variant of PopZ or McdB.

As mentioned above, our data supports that the *nature* of IbpA association between condensates and a well-established solid aggregate is different. This claim certainly still holds for cl⁹⁹, but we have softened the language throughout the manuscript regarding the phenomenon occurring with other complexes and chaperones. As we speculate in the paper, the difference is likely biophysical in nature – condensates are more amenable to penetration compared to solid aggregates. In any case, we, nor the original developers of the PopTag, have yet to engineer aggregation-prone variants of the PopTag or McdB. We agree that going forward, several follow-up experiments are needed for us to better understand how protein complexes, of varying material state, differentially interact with the fold-or-destroy triage system of bacterial cells, specifically their sets of chaperones and even their proteases.

Minor Points:

Fig 2 panels B vs C: At 4h the percentage of cells with a focus in panel B is ~50% for McdB, but density of points in panel C is nowhere near 50% with foci, 50% without. Instead, 170 cells have a focus, and 834 cells do not. If the ratio is actually closer to 50/50, why were only ¼ as many cells with foci detected/quantified?

As described in the methods, the experiment for the data presented in Fig. 2d was done under different induction conditions than that of Fig. 2a-c. The goal was to induce expression to a point where the population has cells with and without a focus, to approximately 50%. During our reanalysis of the data, motivated by these reviews, we noticed some inaccuracies in our automated focus detection and classification process. We updated our analysis script to improve the handling of false positives and negatives. Specifically, we added a filtering step that adds focus intensity as a metric. Thank you for noticing the discrepancy.

E. coli as a model system: the authors should more clearly motivate using *E. coli* as the model 'test tube' for expressing heterologous proteins from species separated by hundreds of millions of years of evolution, and characterizing the resulting condensates. I don't disagree with this choice, but it is important to explicitly define why it is informative to use *E. coli* instead of the protein's native organism, or a eukaryotic system like yeast, where the diffraction limit would be less of an issue.

We agree that we did not provide a sufficient explanation for why *E. coli* is the best choice for these studies. We have added the following text to the introduction:

E. coli as the model system provides a wealth of molecular biology tools. It was also chosen over larger eukaryotic cells, such as yeast, because phase separation is influenced by crowding, and we required a cytoplasm representative of the environment for bacterial proteins. Finally, E. coli is also a pragmatic choice for the study of heterologously expressed bacterial proteins because it prevents associations with potential binding partners in the native host. For example, as noted above, in the native organism S. elongatus, McdB associates with carboxysomes, which complicates a reductionistic study of McdB phase separation in vivo.

Line 47: Introduce more about McdB: the specific organism it is from, which is not mentioned, its properties that make it an interesting condensate-forming protein to study, and in both the introduction/discussion, point out the significance of the measurements made in this manuscript to broader questions about the function of McdB. This connection is missing and important. The same could be said for the PopTag 'controls:' did we learn anything new that is functionally relevant for PopZ's function in *C. crescentus*? Or just confirm our existing understanding? Or raise new questions?

We agree that we did not provide sufficient background and rationale for the study of McdB in this paper. We now provide the following in the introduction:

We specifically examined Maintenance of carboxysome distribution protein B (McdB) from the cyanobacterium, Synechococcus elongatus PCC 7942. Carboxysomes are carbon-fixing protein-based organelles that are subcellularly distributed in cyanobacterial cells as well as in some chemoautotrophs. McdB is part of a two-protein system responsible for spatially organizing carboxysomes in the cell. McdB associates with carboxysomes in a currently unknown manner, and acts as an adaptor, linking the carboxysome cargo to its positioning ATPase, called McdA. McdA forms dynamic gradients on the nucleoid in response to McdB-bound carboxysomes, distributing them across the length of the

nucleoid. McdB robustly forms phase-separated droplets in vitro, and mutants that are unable to form condensates in vitro are also defective in carboxysome positioning in vivo (23–25). It remains to be determined if McdB forms condensates in its native host because McdB associates with carboxysomes, making it difficult to parse this association from its potential phase separation activity in vivo.

Throughout the paper, we also now make connections between our findings with the PopTag, its use as a SynBio tool for the study of condensates, and functional relevance with the PopZ micro domain.

Line 179-180: Confusing wording: “expression was stopped to maintain a constant cellular protein level.” It is not a constant level, the point is that removing the inducer but allowing growth to continue means the protein level will drop, as is written in the next line.

The reviewer is correct. We now replace “expression was stopped” with “the inducer was removed from the media”.

Line 142 and discussion: A c_{sat_app} in vivo is estimated, but the significance of the value is not discussed relative to what is known, and without this, it is not a particularly meaningful measurement. Some context that might make these measurements more meaningful would be:

(1) comparison with the cellular concentration of PopZ and McdB in vivo in their native organisms (and the authors can presumably do this for McdB, since they are pioneering this system). If the apparent c_{sat} from *E. coli* is quite different from this, it would be interesting and useful to speculate why – due to differences in the subcellular interaction partners in the native system? Due to other conditions in the native host environment?

(2) Comparison with in vitro measurements

Thank you for the suggestion, we have added the following to address these missing points:

*Full-length PopZ has a similar linker length as our PopTag^{LL} fusion. In vitro, PopZ forms condensates at concentrations as low as $1 \mu M^{26}$. However, PopZ micro domains in its native host, *Caulobacter crescentus*, were estimated to be $\sim 50 \mu M$ at the cell poles. Therefore, our in vivo c_{sat_app} of PopTag^{LL} is consistent with previously reported in vivo values^{17, 26}. As for McdB, its cellular concentration in its native host, *Synechococcus elongatus*, has yet to be determined. In vitro, we have shown that McdB can form condensates at concentrations as low as $2 \mu M^{25}$. Here we report estimates several fold higher than the in vitro c_{sat} . However, given that McdB associates with carboxysomes in *S. elongatus*, we speculate that this association significantly drops the c_{sat} of McdB, which likely explains this discrepancy.*

Reviewer #2 (Remarks to the Author):

The understanding of biomolecular condensates in bacteria is physiologically important but limited in technology. In this study, an experimental framework was developed to assess whether a protein focus could be determined as a phase-separated condensate. The formation, reversibility, and dynamics of condensate-forming proteins were tested in *Escherichia coli* by c_{sat} calculation, changes in cell shape or temperature, FRAP, and single-molecule tracking. The topic is no doubt interesting and important. However, there are still some major issues which need to be addressed. For example, some conclusions

may not be supported by the results or being overstated in the manuscript. Some data interpretations of methods need to be cautious. Please see below for more details.

1. The results of Fig S1 are lacking of quantifications, including those of cell curvature localizations, temperature shifts and drug-induced changes of mNG-McdB.

Quantifications are now provided. Furthermore, we now simply note the intriguing observation of cell curvature induced by the condensate; speculation regarding the induction of cell curvature has been removed (ln99-104).

“We speculate mNG-McdB foci wet to the membrane via nonspecific electrostatic associations and locally occlude cell wall synthesis.” Any evidence for this claim? What activity of McdB could cause the local changes in cell wall? Please clarify this.

We now simply note the intriguing observation of cell curvature induced by the condensate. Speculation regarding the induction of cell curvature has been removed (ln99-104).

“The resulting asymmetry in cell wall growth thereby induces cell curvature.” Any evidence for the asymmetry in cell wall growth?

The speculation has been removed (ln99-104).

2. As stated by the authors that the temperature shifts and drug-induced changes to cell volume may involve pleiotropic effects, the claim of “suggesting that the cellular levels of mNG-McdB dropped below csat” needs to change to “indicating that the cellular levels of mNG-McdB may drop below csat”.

We have softened this language as suggested throughout the manuscript.

Fig S1f, if it is possible that the shift of fluorescent signal to the opposite end is the result of protein de-novo synthesis, rather than the solubilization of the opposing focus? Please quantify the results and explain this.

The localized cell lysis technique immediately kills the cell. Therefore, little to no additional protein expression is expected. We also now add cl^{a99} as a control (Fig. S1e) and provide quantification for comparison with McdB (Fig. S1f).

3. What's the difference between mCherry-mcdB (Fig 2) and mNG-McdB (Fig S1) constructions? Why the authors emphasize that mCherry-mcdB (Fig 2) is the tunable expression?

The photoactivatable fluorescent proteins (PAmCherry fused proteins) in Fig 2 could be tunable constructions. However, only the turn on of expression but no turn off of expression was shown. Hence, no big difference was demonstrated in comparison to the IPTG induction.

We apologize for the misunderstanding. We have modified our descriptions of the two expression systems as suggested by Reviewer 1.

The mCherry-mcdB (Fig 2) and mNG-McdB (Fig S1) fusions are under the control of the P_{trc99A} and T7 promoters respectively. The T7 promoter induces a switch-like protein expression, while the P_{trc99A} provides single-cell tunable protein expression.

The photoactivatable fluorescent proteins (PAmCherry fused proteins) in Fig 2 is under the P_{trc99A} promoter, which provides single-cell tunable protein expression similar to the mCherry-mcdB (Fig. 2). Expression of mCherry-mcdB and PAmCherry were 'turned off' by removing the inducers from the media (i.e. replacing the current medium with fresh medium without inducers in Fig 4).

4. Why choose the five proteins of clagg, PopTag^{SL}, PopTag^{LL}, McdB, and McdB^{sol} to demonstrate the framework in this study? Are they being selected randomly? What's the logic behind these five proteins? In addition, the authors should describe the PopTag more specifically, so does the linker for PopTag^{LL}. Is PopTag^{LL} using PopZ sequence as the protein linker?

We apologize for the lack of clarity in describing why this protein set was chosen. McdB is our protein of interest, which phase separates *in vitro* and we want to determine if it phase separates *in vivo*. We now provide an introduction to McdB and the rationale for why it was chosen as our protein of interest in the introduction. We apologize for this omission.

We specifically examined Maintenance of carboxysome distribution protein B (McdB) from the cyanobacterium Synechococcus elongatus PCC 7942. Carboxysomes are carbon-fixing protein-based organelles that are subcellularly distributed in cyanobacterial cells as well as in some chemoautotrophs. McdB is part of a two-protein system responsible for spatially organizing carboxysomes in the cell. McdB associates with carboxysomes in a currently unknown manner, and acts as an adaptor, linking the carboxysome cargo to its positioning ATPase, called McdA. McdA forms dynamic gradients on the nucleoid in response to McdB-bound carboxysomes, distributing them across the length of the nucleoid. McdB robustly forms phase-separated droplets in vitro, and mutants that are unable to form condensates in vitro are also defective in carboxysome positioning in vivo²³⁻²⁵. It remains to be determined if McdB forms condensates in its native host because McdB associates with carboxysomes, making it difficult to parse this association from its potential phase separation activity in vivo.

cl^{agg}, PopTag^{SL}, and PopTag^{LL} were chosen as controls to represent protein assemblies with different properties that are already well-established in the literature. cl^{agg} is known to form insoluble aggregates, PopTag was recently shown to induce phase separation with linker-length dependent condensate properties. The short linker bestows a more viscous state, while the longer linker is expected to behave more 'liquid-like'. PopTag^{SL} uses a short GSGSGS linker while PopTag^{LL} uses a longer 78-aa linker that is native to PopZ protein. We have made modifications to clarify our rationale for the chosen protein set used.

We next determined the concentration dependence of focus formation among our protein set fused to mCherry. Cells were first categorized based on the presence or absence of a focus. We then plotted the fluorescence concentration of these cell types over the time course of protein expression (Fig. 2c). For cl^{agg}, cells lacking a focus displayed little to no fluorescence, while cells with a focus displayed a linear increase in concentration over time. The pattern for PopTag^{SL} was similar to that of cl^{agg}, consistent with the previously established gel-like nature of PopTag^{SL} condensates. In contrast, both PopTag^{LL} and McdB displayed a relatively sharp fluorescence concentration boundary for focus formation. That is, cells without a focus displayed a fluorescence concentration at or below the concentration in cells displaying a focus. McdB^{sol} foci also exhibited this concentration threshold, albeit foci were only observed at significantly higher concentrations five hours post-induction.

5. The main purpose of western blot in Fig S2 should indicate the expression levels of mCherry fusion proteins, instead of the degradation levels of these proteins. Nevertheless, I don't think it's a good idea to use western blot to indicate the degradation levels due to the non-specificity of antibody.

The main point of the western was not to quantify expression levels; we were able to do this more accurately through fluorescence quantification (Fig. 2c,d). However, for this fluorescence quantification to be valid, we needed to ensure there was not a significant level of degradation liberating mCherry from the proteins of interest. The main purpose of the western blot of Fig. S2 was thus to verify limited degradation.

None of the mCherry fusion proteins showed significant degradation or cleavage of the mCherry tag compared to mCherry alone when expressed with the pTrc99A promoter (Supplementary Fig. 2). Therefore, mCherry fluorescence served as a reliable measure of protein expression and localization in vivo.

What I understand is that csat_app in Fig 2 is derived from Fig S4 quantification. However, what's the quantitative signal used in Fig S4 and how to transform this signal into protein concentration. Please clarify this.

For clarification, please refer to the “**Measuring single-cell protein concentrations**” section under Methods. Briefly, to convert intensity counts to photons, we calibrated our EMCCD to obtain the conversion gain (i.e., the number of photoelectrons per intensity count) (Fig. S4a) and the EM gain (the number of electrons per photoelectron) (Fig. S4b). In these measurements, intensity counts are the quantitative signal. These values were subsequently used to determine the number of photons per single-molecule localization (Fig. S4c) and the number of photons per cell under our imaging conditions. We measured 91.1 photons per single-molecule localization and subsequently used this value to determine the number of molecules per cell. We then estimated the cell volume using cell length and width measurements from the phase contrast images. In combination with the number of molecules per cells, these values were then used to calculate molarity.

6. “Intriguingly, PAmCherry-McdB^{sol} also formed high-density regions at the poles, consistent with bulk fluorescence measurements (see Fig. 2a).” Why their localization patterns (in Fig. 2a and Fig. 2f) are different?

We agree with the reviewer's point that our image in Fig. 2a was not representative of the population. We have now changed the image in Fig. 2a to better represent the localization data shown in Fig. 3c.

“we speculate that the localization pattern of the fluidized PAmCherry-McdB condensate is due to nucleoid exclusion by repulsive electrostatic interactions.” Any evidence for this claim, just because the net negative charge of its IDR was increased? The mCherry alone could be a possible control in Fig 2f.

Fig. S6 and its associated Movie 5 provide evidence for our proposal that mCherry-McdB^{sol} is fluidized, but remains nucleoid-excluded. As the nucleoid compacts following drug treatment, the polar signal of mCherry-McdB^{sol} fills the cytoplasmic space previously occupied by the nucleoid. This filling is immediately in concert with nucleoid compaction. We have now provided quantification of mCherry-McdB^{sol} instantaneously filling regions of the cell previously occupied by the nucleoid prior to nucleoid compaction via drug treatment (Fig. S6b,d) as suggested by the reviewer. In contrast, the mCherry-McdB foci remain unchanged after the nucleoid compacts. We do observe a slight rearrangement of the cytoplasmic fraction of McdB, further supporting our statement that the rearrangement of fluidized McdB is likely due to repulsion from the nucleoid.

7. Fig S6, a quantification could be performed to support the conclusion here.

Profile lines for representative cells are now provided in Fig. S6b and d.

8. The interpretation of Fig 3 need to be more cautious because it has a very complicate environment in living cells. The authors did not take the protein degradation or expression leaky into consideration. Also, the authors need to be careful about the photobleaching during the time lapse imaging.

We agree with the sentiments expressed by the reviewer. As suggested by Reviewer 1, we have now added a paragraph highlighting the fact that the bacterial cytoplasm is inhomogeneous and that the cell poles have distinct biophysical properties, including attenuated diffusion (Smigiel WM et al., 2022, PMID: 35960807). This is the main reason we focus on *relative* comparisons of a protein set fused to the same fluorescent protein, all localized to the cell poles, and all exposed to same—albeit complicated—cellular environment. Please note that we do take into account protein degradation, as quantified via westerns in Fig. S2.

A recent study that reports on the inhomogeneity of the bacterial cytoplasm finds slowed apparent protein diffusion at the cell poles compared to other regions of the E. coli cell³⁴. The finding highlights the importance of comparative studies of in-focus diffusion measurements for foci occupying the same cellular location. It is likely that slowed protein diffusion contributes to condensate formation at cell poles. But, even if diffusion is slowed at the poles to some extent, our in-focus diffusion measurements are much slower and non-uniform across our protein set; supporting the proposal that interactions involved in condensate formation further affect protein diffusion.

9. “As the cell divides, the volume decreases, leading to an increase in the concentration back above c_{sat} in the daughter cell, which ultimately results in the reformation of a focus.” The interpretation for Fig S8 makes me confused, since the protein expression was stopped and the concentration will not be increased after cell division.

We agree our previous explanation lacked clarity. Some of the confusion stems from our inappropriate use of the phrase “expression was stopped”. As suggested by Reviewer 1, this phrase has now been replaced with “inducer was removed” at the indicated times. This is an important distinction because, even after the removal of inducer, some residual low-level expression is still expected. Also, as suggested by Reviewer 1, we now quantify our images showing focus reformation (Fig. 4 g-j). The data supports our hypothesis that this phenomenon is due to an asymmetric inheritance of protein upon cell division, which drove the concentration in one daughter cell above c_{sat} .

Intriguingly, some cells reformed McdB and PopTag^{SL} foci immediately after cell division (Fig. 4g-j). We hypothesized that this phenomenon was due to an asymmetric inheritance of protein upon cell division, which drove the concentration in one daughter cell above c_{sat} . Analysis of several division events for cells expressing McdB and PopTag^{SL} (see Methods) confirmed our hypothesis: most daughter cells that formed foci inherit more fluorescence signal than to the other corresponding daughter cell (Fig 4h,j).

10. The results in fig 4 are more reliable. Could the authors provide more details to support the accessibility of this method in the framework? For example, how many molecules was used to probe in fig 4c? What is the variance distribution in this study?

The numbers of trajectories used for each protein condition are shown in Fig. S8, in which we also demonstrate the diffusion coefficient distributions. Three independent single-molecule tracking experiments were done to assemble these distributions. Below we provide the distributions for each replicate, showing great consistency across all experiments.

11. It has been demonstrated that IbpA could serve as a molecular sensor for protein aggregates. However, the penetration of condensates by IbpA was not shown until this work. Since people use this method to discriminate the aggregates and functional proteins before, much more evidences should be provided to support this claim. Two proteins are not having statistical significances.

The importance of following up on our IbpA findings was also noted by Reviewer 1 and by several prokaryotic cell biologists to whom we have presented this work. In our opinion and others, these initial findings with IbpA open the door to a new sub-field only recently being addressed in eukaryotic cells – “Condensostasis” (Ali et al, Nat Cell Biol, 2023). We agree that, going forward, several follow-up experiments are needed for us to better understand how protein complexes and varying material state differentially interact with the fold-or-destroy triage system of bacterial cells, specifically their sets of chaperones and even their proteases.

12. No text lines.

Text lines have now been added. Our apologies.

Reviewers' Comments:

Reviewer #1:

Remarks to the Author:

Overall Review:

The additional experiments and additional analysis of existing data significantly improve the rigor of this study and enhance the study's significance as a suite of quantitative tools to study bacterial condensates using *E. coli* as a chassis. Most of my major comments from the first round of reviews have been addressed. The additional analysis of the existing data in Figure 2 is very helpful. However, from this new analysis, there is one key point of clarification that I believe is critical for the authors' interpretation of concentration dependence *in vivo*:

In the epifluorescence/sCMOS/LED illumination set of experiments shown in Figure 2C, it appears that for both the PopTag^{LL} and McdB constructs, there is a clear difference in concentration for cells with and without a focus, consistent with the authors' model that condensates form above a threshold concentration. However, in the set of experiments using the single-molecule setup/EMCCD/561 laser illumination in Figure 2D, with just the 4h datapoint, it would appear to me that there is no significant difference between the concentration of protein in cells with a focus compared to cells without a focus, for either PopTag^{LL} or McdB. Plotting the single cell data for the 4h timepoint in C in the same manner as shown in D (though the y-axis will be different), along with the same statistical test, and/or some clarification of this difference is needed.

Aside from this key point, the other additional experiments and analysis all significantly strengthen the study. In particular:

- The quantification of single cell fluorescence concentrations for focus dissolution during outgrowth (Fig 4a-f)
- The quantification of asymmetric fluorescence intensity in daughter cells in the case of focus reformation events.
- The new temperature shift experiments with the entire set of mCherry constructs used in the rest of the study (Fig 4k and i).

Minor comments:

- The description of the analysis performed in Fig 4c and f in the caption and text is not clear. My understanding of the purpose of these plots is to address whether focus dissolution during dilution and outgrowth after the inducer is removed occurs at the same concentration as focus formation during induction. Are 'pre' and 'post' points referring to a single frame before and after focus dissolution for a given focus in a movie? This is my interpretation/what would make sense to me, but some elaboration in the caption and/or text would help readers.
- N (number of cells and replicates) is not provided for all figure panels, and in some cases it is not clear when datasets are pooled from multiple biological replicates and also how many independent biological replicates were performed. For example, it is clear that panel b represents 3 biological replicates, but less clear what is being plotted in c (pooled data from all sets?). In (d), it is unclear again if these data are pooled individual cells from all replicates, as the legend says the statistical test was done on the 3 replicates. Consistency in the quantification being performed in panels c and d is important to address my larger concerns about correlating the results from the different datasets in Fig 2C and D.

Reviewer #2:

Remarks to the Author:

This revision provided by Y Hoang et al. has been substantially improved by adding the quantifications and controls throughout the paper. Also, the statements were adjusted by either

removing the conclusion or soften the language.

The understanding of biomolecular condensates is limited in technology in small bacterial cells. Therefore, this paper set out to develop an experimental framework to assess in-vivo biomolecular condensates has great physiological significance. Nevertheless, the current manuscript was not organized well by following this thread of logic because it took a very lot of efforts (four paragraphs) to describe how McdB condensate was forming and how the results were consistent to each other. In contrast, the authors said little about how to use these technologies or experiments to demonstrate a promising biomolecular condensate in cells. Is there a good standard or standard sets could be provided after this research? During the process, which experiment is necessary and which one is not? Which technology is the primary suggestion for the detection of formation, reversibility, or dynamics of condensates? This is my major concern regarding the novelty and the contribution to the field by this great work. The authors need to reorganize the manuscript and discuss more after the research.

Other specific comments:

1, line 134, why emphasize "single-cell"? this "single-cell" is not the traditional "single-cell" people usually saying.

2, line 161, please change to "the latter of which".

3, line 164-166, here in the fig s2 showed the cleavage but not the degradation of the protein. For the degradation itself, authors need to do a time-course monitoring in vitro.

4, line 217-218, the 89-121 μM of csat_app for mCherry-McdB is super high. The authors speculates that the association with carboxysomes will significantly drops the csat of McdB. The increase of local concentration of McdB by association with carboxysomes may be more reasonable.

5, line 253, changes in cellular concentration, osmolarity, and temperature

6, line 316-318, "increasing the temperature of cells with a preformed focus would lead to the dissolution of condensates" only be appropriate for the UCST polymers, but not for the LCST polymers.

7, line 413-414, the findings suggest a different mode of colocalization of IbpA with biomolecular condensates PopTags and McdB versus aggregates cIagg. We still need to be cautious here to avoid overstated here.

8, line 424-425, no evidence for the "potentially other chaperones", could be removed.

9, line 623, cI78EP8

10, PopTagLL uses a longer 78-aa linker that is from PopZ protein, please point out the specific location in PopZ or provide the sequence.

Figures:

1, figure 1, the use of IbpA patterns as a general method to discriminate aggregates and condensates need to be cautious. More evidences should be provided to support this general claim.

2, figure 4b, the labels on y-axis were incomplete; Figure 4c,f, should mention the biological mean of the calculation data for pre- and post- in the text; Figure 4h,j, how many cells were calculated?; line 1113, change j to l; Figure 4l, need a control for each protein without the treatment.

3, figure 6, it seems that the IbpA expression was pretty low. If it is possible that the interaction

pattern be changed when using a higher expressed IbpA in these experiments?

REVIEWER COMMENTS

Reviewer #1 (Remarks to the Author):

Overall Review:

The additional experiments and additional analysis of existing data significantly improve the rigor of this study and enhance the study's significance as a suite of quantitative tools to study bacterial condensates using *E. coli* as a chassis. Most of my major comments from the first round of reviews have been addressed. The additional analysis of the existing data in Figure 2 is very helpful. However, from this new analysis, there is one key point of clarification that I believe is critical for the authors' interpretation of concentration dependence *in vivo*:

In the epifluorescence/sCMOS/LED illumination set of experiments shown in Figure 2C, it appears that for both the PopTagLL and McdB constructs, there is a clear difference in concentration for cells with and without a focus, consistent with the authors' model that condensates form above a threshold concentration. However, in the set of experiments using the single-molecule setup/EMCCD/561 laser illumination in Figure 2D, with just the 4h datapoint, it would appear to me that there is no significant difference between the concentration of protein in cells with a focus compared to cells without a focus, for either PopTag^{LL} or McdB. Plotting the single cell data for the 4h timepoint in C in the same manner as shown in D (though the y-axis will be different), along with the same statistical test, and/or some clarification of this difference is needed.

Thank you for your comment. Regarding the data presentation in this figure, we have updated Fig. 2c to indicate the same estimate of precision as Fig. 2b (the shading in both plots now indicates the standard deviation). To address the concern of significant differences across the two different experiment types, we want to clarify that the data for Fig. 2c was collected using different induction conditions than in Fig. 2d. The experiment in Fig. 2C was performed with 200 μ M IPTG induction, while the experiment in Fig. 2D was performed with 100 μ M and 1 mM IPTG for PopTagLL and McdB, respectively (Methods). Therefore, the 4h time point in Fig. 2c does not correspond the data presented in Fig. 2d. Indeed, in Fig. 2c, there's a statistically significant difference between the cells with a focus and no focus in PopTagLL and McdB across the 2-3 h and 2-5 h time points, respectively, when the statistical test (Welch's t-test) is done on the mean of the three biological replicate means (see Table 1).

Table 1	PopTagLL 2h	PopTagLL 3h	McdB 2h	McdB 3h	McdB 4h	McdB 5h
p-value	0.001	0.006	0.037	1.5e-05	0.006	0.002

In Fig. 2d, the data points are pooled across three independent experiments performed on different days, but the reported values are the mean and standard deviation of the 3 means for each experiment. In this case, there is no statistically significant difference between the cells with a focus and no focus (Table 2). Because of potential variability day-to-day in the laser illumination as well as IPTG induction, we also performed the statistical test on each independent experiment (Table 3 and figure). In this case, all

three McdB replicates exhibit a statistically significant difference between the cells with a focus and no focus. On the other hand, PopTagLL still exhibits no difference between the sets. We hypothesize that under these induction conditions, there is not a significant difference in signal between the cells that do and do not have a detected focus because the cells are close to the c_{sat} point. For consistency, the plot in Figure 2d is now shown as a SuperPlot - color-coded by replicate and showing all data points and the means of each replicate accordingly.

Table 2	PopTag ^{LL} (n=3)	McdB (n=3)
p-value	0.546	0.095

Table 3	PopTag ^{LL} rep 1	PopTag ^{LL} rep 2	PopTag ^{LL} rep3	McdB rep 1	McdB rep 2	McdB rep 3
p-value	0.155	0.016	0.731	1.70E-05	0.001	0.007

Aside from this key point, the other additional experiments and analysis all significantly strengthen the study. In particular:

- The quantification of single cell fluorescence concentrations for focus dissolution during outgrowth (Fig 4a-f)
- The quantification of asymmetric fluorescence intensity in daughter cells in the case of focus reformation events.
- The new temperature shift experiments with the entire set of mCherry constructs used in the rest of the study (Fig 4k and i).

We thank the reviewer for their incredibly thoughtful and thorough comments.

Minor comments:

- The description of the analysis performed in Fig 4c and f in the caption and text is not clear. My understanding of the purpose of these plots is to address whether focus dissolution during dilution and outgrowth after the inducer is removed occurs at the same concentration as focus formation during induction. Are 'pre' and 'post' points referring to a single frame before and after focus dissolution for a given focus in a movie? This is my interpretation/what would make sense to me, but some elaboration in the caption and/or text would help readers.

Yes, your interpretation is correct and we have now added a more descriptive legend on Fig. 4c and 4f to clarify that the data represents the fluorescence concentration of cells pre- and post- condensate dissolution. In the text, we included the sentence below to clarify the meaning of this analysis.

“Furthermore, we measured the fluorescence concentration of cells that exhibited focus dissolution at the frames immediately preceding and immediately following the dissolution event (Fig. 4c, f).”

- N (number of cells and replicates) is not provided for all figure panels, and in some cases it is not clear when datasets are pooled from multiple biological replicates and also how many independent biological replicates were performed. For example, it is clear that panel b represents 3 biological replicates, but less clear what is being plotted in c (pooled data from all sets?). In (d), it is unclear again if these data are pooled individual cells from all replicates, as the legend says the statistical test was done on the 3 replicates. Consistency in the quantification being performed in panels c and d is important to address my larger concerns about correlating the results from the different datasets in Fig 2C and D.

Thank you for the comment. Indeed, the data presented in Fig. 2c is pooled from all 3 biological replicates. For consistency, we've changed that panel to indicate the same estimate of precision as Fig. 2b: in Figs. 2b and 2c, the mean and standard deviation of the means of 3 biological replicates are now plotted. The figure caption has been updated accordingly. In Figure 2d, the data points were pooled from 3 replicates, but the statistical test was done on the means of the 3 replicates. We have updated the plot for consistency: Figure 2d is now a SuperPlot - color-coded by replicate and showing all data points and the means of each replicate accordingly.

Reviewer #2 (Remarks to the Author):

This revision provided by Y Hoang et al. has been substantially improved by adding the quantifications and controls throughout the paper. Also, the statements were adjusted by either removing the conclusion or soften the language.

The understanding of biomolecular condensates is limited in technology in small bacterial cells. Therefore, this paper set out to develop an experimental framework to assess in-vivo biomolecular condensates has great physiological significance. Nevertheless, the current manuscript was not organized well by following this thread of logic because it took a very lot of efforts (four paragraphs) to describe how McdB condensate was forming and how the results were consistent to each other. In contrast, the authors said little about how to use these technologies or experiments to demonstrate a promising biomolecular condensate in cells. Is there a good standard or standard sets could be provided after this research? During the process, which experiment is necessary and which one is not? Which technology is the primary suggestion for the detection of formation, reversibility, or dynamics of condensates? This is my major concern regarding the novelty and the contribution to the filed by this great work. The authors need to reorganize the manuscript and discuss more after the research.

Thank you for sharing this concern regarding the lack of discussion relating to the novelty and contribution to the field this methodology provides. We now add the following addition to the discussion section, which addresses the questions put forth by the reviewer:

“When studying a new condensate, it is crucial to first determine the conditions under which it forms or dissolves. Inducer-controlled protein expression combined with quantitative fluorescence microscopy demonstrated general applicability in identifying the *in vivo* c_{sat} for condensate formation. Our results also indicate that *in vivo* c_{sat} may vary within the cell population. Another critical metric is to determine the reversibility of a focus, as it is one of the hallmarks that distinguish condensates from aggregates. Our approaches for decreasing cellular protein levels below c_{sat} , via changes in cell shape, osmolarity, temperature, or generational dilution, provide accessible methods to probe the reversibility of condensate formation.

The dynamics of proteins inside a focus should then be determined by FRAP and single-molecule tracking. While FRAP is more accessible, it is challenging to collect data from a large number of cells. Single-molecule tracking experiments, on the other hand, enable the assessment of a large sample size and more in-depth analysis of protein dynamics within both the cytoplasm and the focus. However, the material state of a focus cannot be determined based on the diffusion coefficient alone. Even when the diffusion coefficient is relatively low, focus formation can still be reversible, as observed with the gel-like control protein in our study. Therefore, a combination of reversibility and dynamic assessments is essential to determine the material state of a focus in bacterial cells.”

Other specific comments:

1, line 134, why emphasize “single-cell”? this “single-cell” is not the traditional “single-cell” people usually saying.

This wording was a suggestion put forth by reviewer 1 in the first review, which we agreed with. We want to emphasize the differences between the two expression systems that are described in the manuscript: while expression from the T7 promoter yields a very high amount of protein without tunability or controllability, the expression by the P_{trc} promoter is tunable and can be halted by removal of the inducer and addition of glucose. In the previous revision, reviewer 1 suggested that we use “switch-like expression” and “single-cell tunable expression” to describe the overexpression by the T7 promoter and the expression by the P_{trc} promoter, respectively. As a compromise between the different suggestions by the two reviewers, we now add “tunable expression” to better describe the expression with P_{trc} promoter to avoid confusion with the common usage of the term “single-cell”.

2, line 161, please change to “the latter of which”.

“the latter of which” has been added to line 161.

3, line 164-166, here in the fig s2 showed the cleavage but not the degradation of the protein. For the degradation itself, authors need to do a time-course monitoring in vitro. We have changed fig S2 to “cleavage level”.

4, line 217-218, the 89-121 μM of c_{sat_app} for mCherry-McdB is super high. The authors speculates that the association with carboxysomes will significantly drop the c_{sat} of McdB. The increase of local concentration of McdB by association with carboxysomes may be more reasonable.

We have changed lines 217-218 to “increases of local concentration of McdB and/or drops the c_{sat} of McdB”.

5, line 253, changes in cellular concentration, osmolarity, and temperature

The change has been made to line 253.

6, line 316-318, “increasing the temperature of cells with a preformed focus would lead to the dissolution of condensates” only be appropriate for the UCST polymers, but not for the LCST polymers.

We have changed the sentence to:

“We hypothesized that increasing the temperature of cells with a preformed focus would influence condensates, while having little to no effect on insoluble aggregates.”

7, line 413-414, the findings suggest a different mode of colocalization of IbpA with biomolecular condensates PopTags and McdB versus aggregates clagg. We still need to be cautious here to avoid overstated here.

We have changed the sentence to: “Together, the findings suggest a different mode of colocalization of IbpA with the examined biomolecular condensates and aggregates.”

8, line 424-425, no evidence for the “potentially other chaperones”, could be removed. The statement has been modified in the following manner:

“These patterns serve as a proof of concept that IbpA, ~~and potentially other chaperones~~, can serve as a reporter capable of differentiating between these macromolecular assemblies *in vivo*. ~~Future studies will determine if other chaperones share this activity.~~”

9, line 623, cI78EP8

Thank you. cI78EP8 has been changed to cI78^{EP8}

10, PopTagLL uses a longer 78-aa linker that is from PopZ protein, please point out the specific location in PopZ or provide the sequence.

The sequence of the 78-aa linker is now provided in the Methods section:

“The linker sequence used for PopTagLL:

DDAPAEPAAEAAPPPPEPEPEPVSFDEVLELTDPIAPEPELPPLETVDIDVYSPPEP
ESEPAYTPPPAAPVFDRD”

Figures:

1, figure 1, the use of IbpA patterns as a general method to discriminate aggregates and condensates need to be cautious. More evidences should be provided to support this general claim.

We agree with the reviewer and dedicate a paragraph in the Discussion to elaborate on the usage and potential of IbpA as a reporter to discriminate aggregates and condensates.

“Inclusion body binding protein A (IbpA) of *E. coli* belongs to the conserved family of ATP-independent small heat shock proteins, well-established in binding protein aggregates and driving them towards reactivation-prone assemblies^{37–39}. As such, we presumed IbpA would serve as a molecular sensor that would selectively associate with protein aggregates, but not condensates. Instead, we found that IbpA surrounded protein aggregates and penetrated condensates. Moreover, the degree to which IbpA colocalized with condensates strongly correlated with increasing fluidity. Consistent with our findings, the Drummond group has recently shown that condensates are dispersed by chaperones far more rapidly than misfolded aggregates⁴⁰. These findings warrant a reevaluation of the function of chaperone systems governing protein homeostasis and demonstrate the utility of IbpA, and potentially other chaperones, as molecular sensors for the material state of fluorescent foci in bacteria.”

2, figure 4b, the labels on y-axis were incomplete; Figure 4c,f, should mention the biological mean of the calculation data for pre- and post- in the text; Figure 4h,j, how many cells were calculated?; line 1113, change j to I; Figure 4l, need a control for each protein without the treatment.

The y-axis of Fig. 4b is now fixed. We also added a more descriptive legend for Fig. 4c and 4f to clarify that the data represents the fluorescence concentration of cells pre- and

post-condensate dissolution. In the text, we included the sentence below to clarify the meaning of this analysis.

“Furthermore, we measured the fluorescence concentration of cells that exhibited focus dissolution at the frames immediately preceding and immediately following the dissolution event (Fig. 4c, f).”

For Fig. 4h and 4j, the plot indicates the number of division events that were analyzed. To further clarify, we've included the number of daughter cells in the Figure 4 caption. Thank you for catching the typo in the caption for Fig. 4l; it has now been changed from 4j to 4l.

Regarding the suggestion of including a control for each protein without the temperature ramp treatment, we do not have a stage-top with refrigeration so we cannot do the suggested experiment of imaging at 4 °C for 30 minutes. However, for this reason, each strain was kept at 4 °C for over 1 hour prior to the temperature ramp and imaging experiments shown.

3, figure 6, it seems that the IbpA expression was pretty low. If it is possible that the interaction pattern be changed when using a higher expressed IbpA in these experiments?

In our experiments, IbpA not regulated by an inducible promoter; the experiment provides native expression levels. Nevertheless, we observed that even after growing the cells with cl^{agg} foci overnight, IbpA still does not completely penetrate the aggregates as shown in the figure below (Bar: 1 μ m).

Reviewer #2 (Remarks on code availability):

I am not the expert in coding.

Data processing and analysis scripts for this study were written in MATLAB and Python. All image analysis scripts, and the SMALL-LABS and NOBIAS algorithm packages are available on GitHub (<https://github.com/BiteenMatlab>).